EMBO
Molecular Medicine

# Functional rescue and AI analysis of a human inactivating GPCR mutation using a small molecule

Debajyoti Das [1], Amanda Wyatt [1], Sarath Sivaprasad[2], Vanessa Wahl [1], Sen Qiao[1], Fabien Ectors [3], Zulfiah M Moosa[4], Claire L Newton [4,5], Mario Fritz[2], Robert P Millar [4,5,6 ✉] & Ulrich Boehm [1 ✉]

## Abstract

G protein-coupled receptors (GPCRs) carry out the majority of cellular transmembrane signaling. Many pathologies have underlying GPCR mutations, most of which cause misfolding and GPCR cell surface trafficking failure. Large libraries of existing small molecule GPCR ligands could be repurposed as pharmacological chaperones (PCs) which restore mutant GPCR folding and function, presenting an exciting alternative to complex gene repair, yet such in vivo studies are limited. Therefore, as proof-of-concept, we use one such known ligand/PC, Org42599/Org43553, to show functional rescue in mice bearing an inactivating human luteinizing hormone receptor (LHR) mutation. Mutant males had delayed puberty and Leydig cell LHR signaling impairment, however, fertility was unaffected. Mutant females had irregular estrous cycles, anovulation, abrogated ovarian LHR signaling, and complete infertility. PC treatment of mutant females restored LH signaling and estrous cyclicity. To characterize treatment efficacy, we developed an AI algorithm that reliably identified inherent differences among experimental groups, enabling functional analysis of the treatment effect in vivo. Our data set the stage to integrate AI analysis with GPCR-targeting PC molecules to treat diverse GPCR-based diseases.

Keywords AI; Calcium Imaging; Inactivating Mutation; Luteinizing Hormone Receptor; Pharmacological Chaperone
Subject Categories Computational Biology; Molecular Biology of Disease; Urogenital System

## Introduction

Mutations resulting in the misfolding of proteins result in diverse diseases. Pharmacological chaperones (PCs) are cell-permeant small-molecule analogs (Bernier et al, 2004; Papp and Csermely, 2006), which engage with misfolded mutant proteins and stabilize their folding, thus allowing them to bypass the scrutiny of the cellular quality control systems and degradative pathways (Conn and Ulloa-Aguirre, 2010; Conn et al, 2007; Radomsky et al, 2024; Ulloa-Aguirre et al, 2021). They can rescue misfolded proteins harboring a range of mutations because they allosterically engage with sites which stabilize the protein rather than the sites of amino acid mutation. A therapeutic alternative is repair of the mutation through gene targeting, a much more complex method in that it requires a specialized procedure for each individual mutation. Moreover, in the case of G protein-coupled receptors (GPCRs), which are the most important drug targets accounting for approximately 40% of Food and Drug Administration-approved drugs (Yang et al, 2024), there is no specific requirement for novel PC development as the pharmaceutical industry has already developed vast numbers of cell-permeant highly specific non-toxic agonists and antagonists which can be repurposed as PCs. Therefore, PCs represent an attractive emerging therapeutic possibility to rescue mutations causing protein misfolding which underlie numerous diseases.

While PCs have been used clinically in diseases such as cystic fibrosis and phenylketonuria (Mijnders et al, 2017; Milla et al, 2017; Vockley et al, 2014), demonstrating the validity of this approach in protein misfolding disorders, none are currently approved for targeting mutations in GPCRs, the largest and most diverse vertebrate protein family consisting of ~800 cell surface receptors which are critical for regulating diverse physiological processes (Hauser et al, 2017; Katritch et al, 2012; Yang et al, 2021). Since most of the amino acids constituting GPCRs are involved in intramolecular interactions important for their three-dimensional configuration, most mutations within the protein-coding region give rise to misfolding of the GPCR protein and a failure of the receptor to traffic to the plasma membrane and the consequential inability to respond to their cognate ligands (Conn and Ulloa-Aguirre, 2010; Newton et al, 2016; Oksche and Rosenthal, 1998; Sahni et al, 2015; Tao, 2006; Ulloa-Aguirre et al, 2014; Ulloa-Aguirre et al, 2021; Ulloa-Aguirre et al, 2022). Inactivating (loss-of-function) mutations of GPCRs are responsible for a large number of pathological conditions, including diabetes insipidus (Bichet,

[1]Department of Pharmacology, Center for Molecular Signaling (PZMS), Center for Gender-Specific Biology and Medicine (CGBM), Saarland University School of Medicine, 66421 Homburg, Germany. [2]CISPA Helmholtz Center for Information Security, 66123 Saarbrücken, Germany. [3]FARAH Mammalian Transgenics Platform, Liège University, 4000 Liège, Belgium. [4]Centre for Neuroendocrinology, Department of Immunology, Faculty of Health Sciences, University of Pretoria, 0084 Pretoria, South Africa. [5]Deanery of Biomedical Sciences, University of Edinburgh, EH8 9XD Edinburgh, UK. [6]Institute for Infectious Diseases and Molecular Medicine, Department of Integrative Biomedical Sciences, Faculty of Health Sciences, University of Cape Town, Observatory, 7925 Cape Town, South Africa. ✉E-mail: bob.millar@up.ac.za; ulrich.boehm@uks.eu

2006), obesity (Tao, 2010), diabetes mellitus (Varney and Benovic, 2024), and reproductive dysfunction (including hyper- and hypogonadotropic hypogonadism) (Francou et al, 2011; Nimri et al, 2011; Schoneberg et al, 2004; Ulloa-Aguirre et al, 2021). A number of studies have demonstrated the capacity of PCs to restore cell surface expression of mutant GPCRs in vitro (Newton et al, 2011; Newton et al, 2021); however, these studies have provided limited mechanistic insight into restoration of signaling by PCs and only very few have been translated into in vivo rescue studies (Ahmed et al, 2019; Bernier et al, 2006; Janovick et al, 2013; Ortega et al, 2022; Rene et al, 2021). This is mainly due to a lack of preclinical models carrying mutations which impair receptor folding and which also provide genetic access to the affected cells, a prerequisite for functional investigation of the pharmacological target at a cellular level. How PCs alter functional signaling and thereby recover receptor function in the affected primary cells in vivo, remains largely unexplored.

To experimentally address these questions, we developed a novel preclinical murine model enabling us to establish proof-of-concept in vivo GPCR rescue studies and to study signaling and PC-driven rescue of GPCR function in primary cells in an intact mouse. We focused on a well-studied GPCR that is essential in the control of fertility by mediating sexually dimorphic luteinizing hormone (LH) action. As the focus of this study, we chose the human T461I mutation in the LH receptor (LHR), which was identified in a patient presenting with Leydig cell hypoplasia Type 1 resulting in infertility (Kossack et al, 2008). Like many other LHR mutations, this mutation impairs trafficking of the receptor to the cell membrane, hindering its binding to the hormone (Newton et al, 2016). The glycoprotein subfamily of receptors, in which the LHR falls, form a unique structurally and functionally distinct subclass of GPCRs for which in vivo rescue is yet to be demonstrated.

Due to the important role of the LHR in reproductive function and the clinical value of targeting this receptor in assisted reproductive technologies and treatment of reproductive dysfunctions, there has been much interest in the development of orally active low molecular weight non-peptide LHR agonists, thus circumventing the need for multiple injections of natural polypeptide hormones. These compounds have been successfully used to induce ovulation in healthy cycling women. One such compound, Org42599/Org43553, which we have designated as LHR-Chap, displays high affinity for the LHR with low affinity for follicle-stimulating hormone (FSH) receptor and, in a clinical study, was able to induce ovulation in GnRH-suppressed, FSH-primed healthy females with no adverse effects (Gerrits et al, 2013). Pertinently, we have previously shown in heterologous cells in vitro that LHR-Chap can also act as a PC and supports trafficking of several mutant LHRs, including T461I, to the cell membrane, restoring responsiveness to LH (Newton et al, 2011; Newton et al, 2021). However, the inherent complexity and dynamicity of the female reproductive endocrine system make in vivo functional rescue of this GPCR particularly challenging in females, and has not yet been attempted using a PC.

Here, we describe the generation of an LhrT465I-IRES-Cre (LhrT465I-IC) mouse line carrying the corresponding murine mutation (i.e., T465I in the terminal exon 11) of the *Lhcgr* gene. There is high homology between the mouse and human LHR protein sequences, and this residue (and the surrounding region) is identical between the two. Furthermore, we have shown in vitro in

heterologous cells that introduction of this mutation into the mouse LHR has a similar effect on cell surface expression as the corresponding mutation (T461I) in the human receptor (albeit to a somewhat lesser extent) and that, importantly, it displays a similar increase in cell surface expression following LHR-Chap incubation (Appendix Fig. S1). In the current study, we demonstrate the utilization of LHR-Chap to "rescue" this inactivating LHR mutation in the mouse model and show that this rescue is capable of restoring normal ovarian functions such as ovulation and formation of *corpora lutea*. By including Cre recombinase alongside the mutation within the recombinant *Lhcgr* allele, we created the model such that it enabled direct genetic access to the LHR-expressing cells. This facilitated functional analysis of signaling within cell populations carrying either mutant or control alleles and exploration into how this is modulated upon PC rescue. Our data demonstrate functional restoration driven by PC rescue of a GPCR in vivo and offer a promising therapeutic possibility for the treatment of human GPCR mutations. To effectively capture the inherent cellular variability and heterogeneity that are hallmarks of primary pharmacological target cells in all biological tissues, we applied artificial intelligence (AI) to develop a preclinical feature-learning classifier that reliably identified the PC-treated group, distinguishing it from the control and mutant groups with excellent accuracy. Strikingly, our AI model detected intrinsic spontaneous functional changes caused by the inactivating human GPCR mutation that could be used to classify mutant cells, precluding the need to rely on exogenous stimulation of the mutant receptor. This showcases the utility and potential preclinical impact of AI in studying the functional effect of inactivating mutations and testing the effectiveness of potential drugs. By providing a more detailed understanding of signaling mechanisms behind PC-based rescue, these data pave the way for in vivo applications of PCs to other GPCR mutations and provide a preclinical basis for exploration of therapeutic possibilities using PCs.

# Results

## Experimental strategy

To introduce the T465I mutation into the mouse genome and make the LHR cells amenable for functional analysis and genetic manipulation, we generated knock-in mice harboring an LhrT465I-IC allele (Fig. EV1A–D). Transcription of the mutant *Lhcgr* allele yields a bicistronic mRNA (Candlish et al, 2015) from which two proteins are independently translated: mutant LhrT465I and Cre recombinase. To visualize LHR-expressing cells in these animals, we crossed heterozygous LhrT465I-IC to eROSA26-τGFP (eR26-GFP) reporter mice (Wen et al, 2011). In the resulting LhrT465I-IC/eR26-GFP animals (Fig. EV2A), Cre-mediated recombination results in GFP labeling in the cells expressing LHR. In the male mice, we detected robust GFP expression in testicular Leydig cells as well as in immature spermatozoa (Fig. EV2B,C). We also observed GFP expression in the mature spermatozoa in the epididymis (Fig. EV2D), indicating LHR expression occurs at this developmental stage. In females, *corpora lutea*, thecal, and interstitial/stromal cells (Fig. EV2E,F), as well as granulosa cells in the ovaries, expressed GFP (Fig. EV2G). These

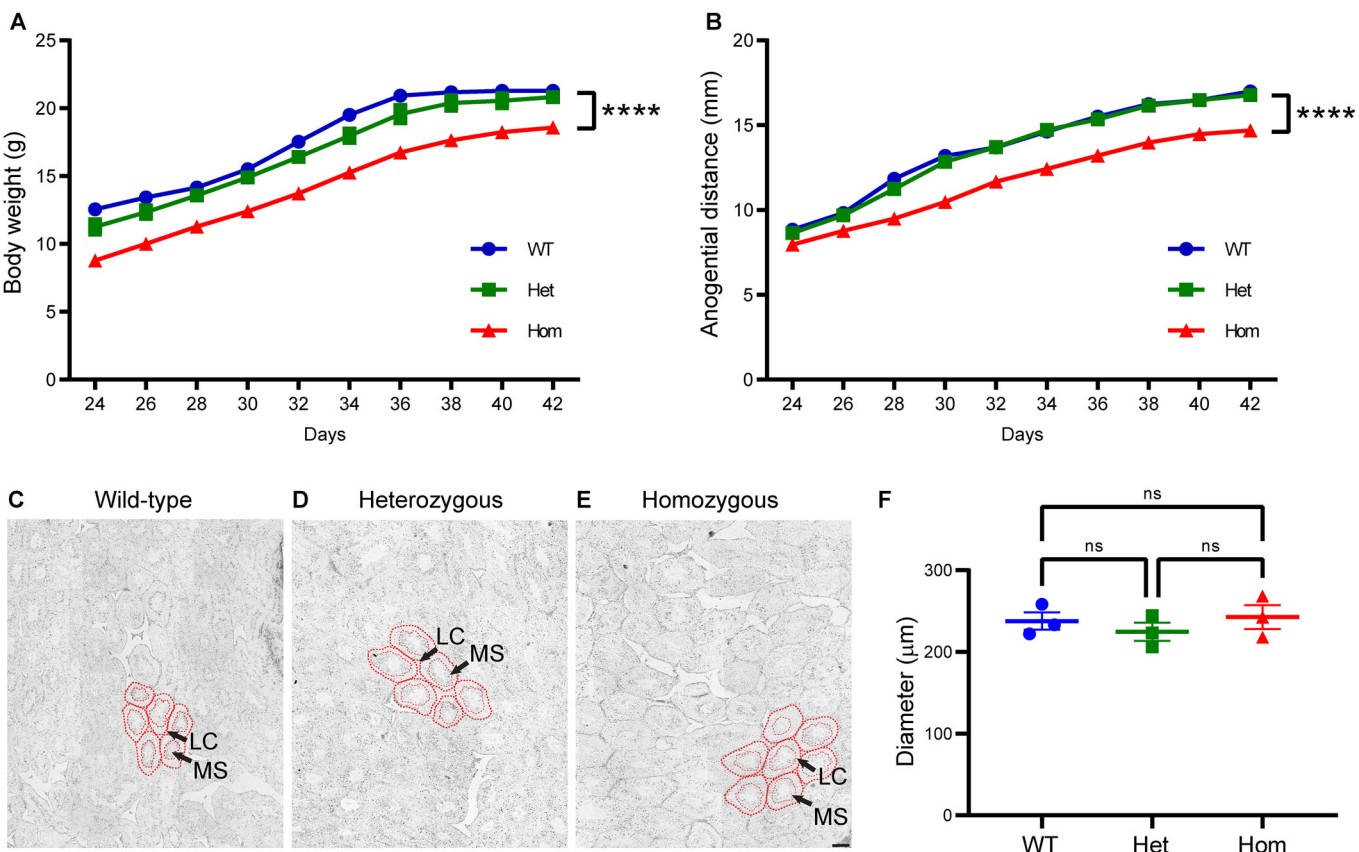

**Figure 1. Reduced body weight and delayed puberty onset in homozygous mutant (LhrT465I-IC⁺/⁺) males.**

In adults, the LhrT465I mutation does not affect testicular histology. (A) Male homozygous (Hom) mutant mice and their heterozygous (LhrT465I-IC⁺/⁻) (Het) and wild-type (LhrT465I-IC⁻/⁻) (WT) littermates were weighed every other day from postnatal day (PND) 24 to PND 42. The body weights of the Hom mice were consistently lower than those of their Het and WT littermates throughout the entire study period. N = 7 (WT), 9 (Het), and 6 (Hom) mice. WT vs. Hom at PND 24 through 32, ****P < 0.0001; PND 34, ***P = 0.0001; and PND 36 through 42, ****P < 0.0001. WT vs. Het at PND 24, P = 0.1502; PND 26, P = 0.1693; PND 28, P = 0.3922; PND 30, P = 0.3355; PND 32, P = 0.0707; PND 34, *P = 0.0496; PND 36, P = 0.1977; PND 38, P = 0.4177; PND 40, P = 0.4204; and PND 42, P = 0.6472. Het vs. Hom at PND 24, **P = 0.006; PND 26, **P = 0.0032; PND 28, ***P = 0.0009; PND 30, ***P = 0.0001; PND 32, ***P = 0.0003; PND 34, **P = 0.0039; PND 36, **P = 0.0089; PND 38, **P = 0.0035; PND 40, **P = 0.0071; and PND 42, **P = 0.0038. (B) Anogenital distances were also measured for these mice groups. Although initially similar, the anogenital distances of the Hom mice were significantly lower than those of their Het and WT littermates for most of the study period. N = 7 (WT), 9 (Het), and 6 (Hom) mice. WT vs. Hom at PND 24, P = 0.1318; PND 26, *P = 0.0448; PND 28, **P = 0.0015; PND 30, ***P = 0.0001; PND 32, ***P = 0.0004; PND 34, **P = 0.0013; PND 36, **P = 0.0013; PND 38, ***P = 0.0003; PND 40, **P = 0.001; and PND 42, ****P < 0.0001. WT vs. Het at PND 24, P = 0.8269; PND 26, P = 0.9024; PND 28, *P = 0.0294; PND 30, P = 0.4611; PND 32, P = 0.9954; PND 34, P = 0.936; PND 36, P = 0.7207; PND 38, P = 0.9339; PND 40, P = 0.9996; and PND 42, P = 0.4696. Het vs. Hom at PND 24, P = 0.1314; PND 26, *P = 0.0277; PND 28, **P = 0.0089; PND 30, ***P = 0.0002; PND 32, ***P = 0.0002; PND 34, ***P = 0.0006; PND 36, **P = 0.0013; PND 38, ***P = 0.0002; PND 40, ***P = 0.0003; and PND 42, ****P < 0.0001. (C–E) Hematoxylin and eosin staining of sections of testes from adult homozygous mutant (LhrT465I-IC⁺/⁺) males and WT (LhrT465I-IC⁻/⁻) and heterozygous (LhrT465I-IC⁺/⁻) littermates did not reveal any differences in seminiferous tubule diameter. LC, Leydig cells; MS, mature spermatozoa. Scale bar: 100 μm. (F) Quantification of seminiferous tubule diameter; no difference was observed among WT, heterozygous (Het), and homozygous (Hom) mutant mice. ns indicates lack of statistical significance (P = 0.742 for WT vs. Het, P = 0.8977 for WT vs. Hom, and P = 0.4946 for Het vs. Hom). Data information: In (A, B, F), data were presented as mean ± SEM. *P ≤ 0.05, **P < 0.01, ***P < 0.001, and ****P < 0.0001 (two-way ANOVA with Tukey's post hoc test for (A, B), one-way ANOVA with Tukey's post hoc test for (F)). Source data are available online for this figure.

observations corroborate the previously reported pattern of gonadal LHR expression (Chen et al, 2000; Lukassen et al, 2018; Su et al, 2023; Yung et al, 2014; Zhang et al, 1994) and demonstrate the validity of our genetic approach.

## Reduced body weight and delayed puberty onset in homozygous mutant males

To test whether the T465I mutation of the LHR affects puberty onset in LhrT465I-IC male mice, we measured body weight and anogenital distance, a marker of puberty and fertility in both mice and humans (Eisenberg and Lipshultz, 2015; Hotchkiss and Vandenbergh, 2005; Thankamony et al, 2016). We found that the body weights of the homozygous (Hom) mutant males (LhrT465I-IC⁺/⁺) were significantly lower than those of the wild-type (WT, LhrT465I-IC⁻/⁻) (Fig. 1A; WT vs. Hom at postnatal day [PND] 24, ****P < 0.0001; WT vs. Hom at PND 42, ****P < 0.0001; two-way ANOVA, Tukey's post hoc test) and heterozygous (Het) mice (LhrT465I-IC⁺/⁻; Fig. 1A; Het vs. Hom at PND 24, **P = 0.006; two-way ANOVA; Het vs. Hom at PND 42, **P = 0.0038; two-way ANOVA) throughout the entire experimental period. Disorders of reproductive function due to LHR mutations are typically

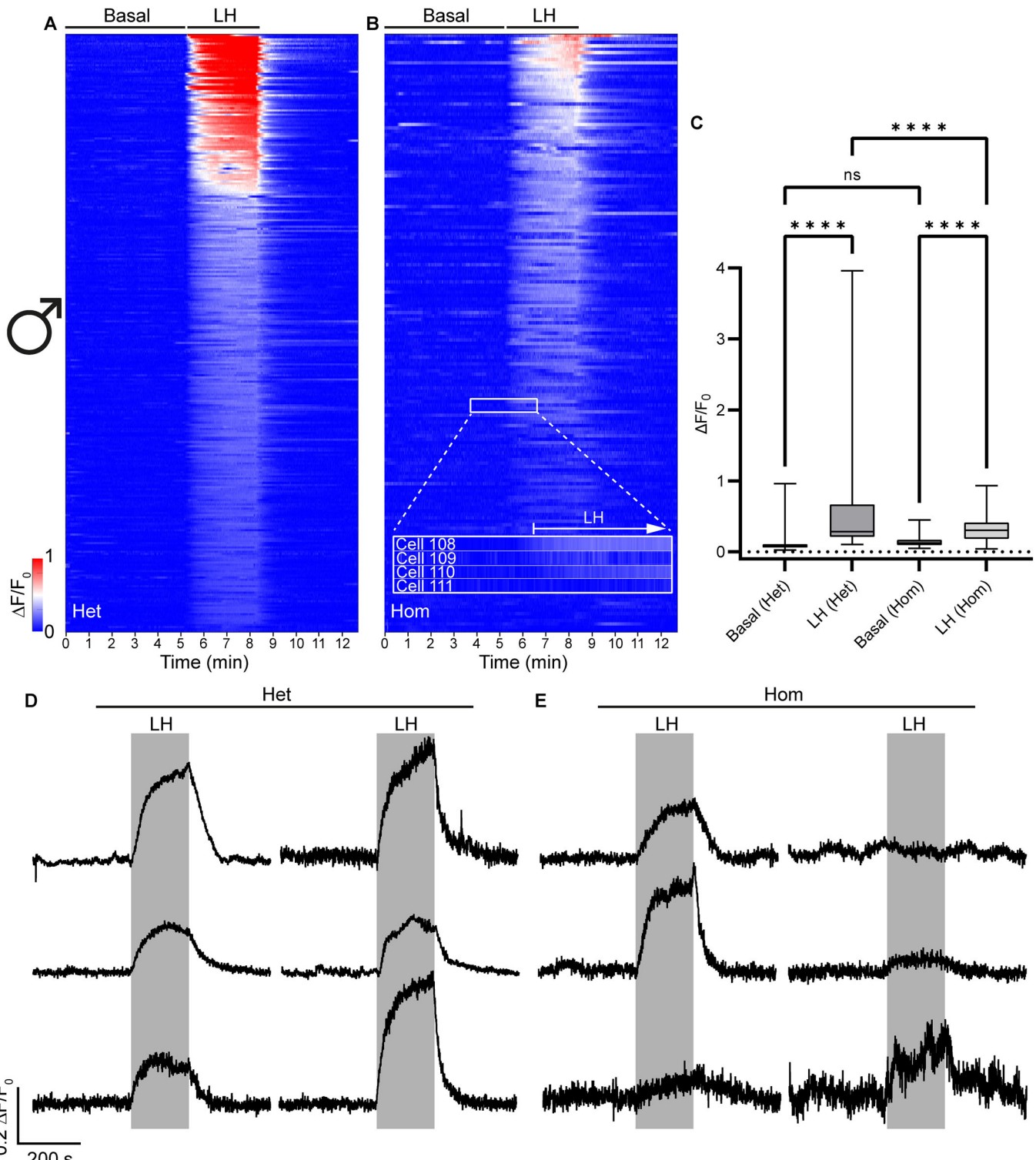

autosomal recessive. In accordance with this, we did not observe a significant difference between the body weights of WT and Het animals (Fig. 1A; WT vs. Het at PND 24, ns $P = 0.1502$; two-way ANOVA). Anogenital distances were also measured at each experimental time-point. No significant differences were observed among the three groups at PND 24 (Fig. 1B; WT vs. Het, ns

$P = 0.8269$; WT vs. Hom, ns $P = 0.1318$; Het vs. Hom, ns $P = 0.1314$; two-way ANOVA). However, at PND 26 and all subsequent time-points, the anogenital distances of the Hom males were significantly lower than those of the WT (Fig. 1B; WT vs. Hom at PND 26, *$P = 0.0448$; WT vs. Hom at PND 42, ****$P < 0.0001$; two-way ANOVA) and heterozygous animals (Fig. 1B; Het vs. Hom at PND

**Figure 2.  Leydig cells in the testes of LhrT465I-IC$^{+/+}$/eR26-GCaMP3$^{+/-}$ mice show diminished responses to LH.**

(A) Heatmap showing collected normalized fluorescence intensities from 278 Leydig cells of LhrT465I-IC$^{+/-}$/eR26-GCaMP3$^{+/-}$ (Het) mice (A, pooled from 3 mice; this is the control group). The heatmaps consist of three sections. The first part indicates the time from the start of the experiment up to LH application. This part of the heatmap shows the spontaneous/basal activity of the Leydig cells. The second part (marked by "LH") indicates the bath application of 1 IU/ml LH. The third part indicates washing of the LH with recording buffer. Cells in the heatmap have been represented in descending order of fluorescence intensities of their Ca$^{2+}$ responses to LH. Scale of the changes in fluorescence intensities in the heatmap are indicated by the color-coded bar. (B) Heatmap showing collected normalized fluorescence intensities from 175 Leydig cells of LhrT465I-IC$^{+/-}$/eR26-GCaMP3$^{+/-}$ (Hom) mice (B, pooled from 3 mice). Inset in (B) magnifies a part of the heatmap to demonstrate calcium response signals from four individual cells (Cells 108–111); this also provides a closer view of the change in fluorescence intensities elicited by LH application. (C) Box-and-whisker plot showing quantification of the total change in fluorescence intensities for all cells before ("Basal") and during LH application, for Het and Hom mice. Asterisks depict statistical significance (****$P < 0.0001$ for Basal [Het] vs. LH [Het], LH [Het] vs. LH [Hom], and Basal [Hom] vs. LH [Hom]), ns ($P = 0.5435$ for Basal [Het] vs. Basal [Hom]) indicates lack of statistical significance. (D, E) Representative traces of 6 Leydig cells each from Het (D) and Hom (E) mice, showing specific response to LH application (indicated by the gray rectangle). Data information: In (C), data were presented as box-and-whisker plots. The boxes themselves span from the first quartile (25%) to the third quartile (75%), representing the interquartile range where the central 50% values lie. Inside each box, a line denotes the median value. The whiskers of each boxplot extend from the ends of the box to the minimum and maximum values. ****$P < 0.0001$ (two-way ANOVA with Bonferroni post hoc test). Source data are available online for this figure.

26, *$P = 0.0277$; Het vs. Hom at PND 42, ****$P < 0.0001$; two-way ANOVA). In contrast, there were no significant differences between the WT and Het animals in anogenital distance at any of the time-points observed (Fig. 1B; WT vs. Het at PND 26, ns $P = 0.9024$; WT vs. Het at PND 42, ns $P = 0.4696$; two-way ANOVA). Taken together, these results indicate incomplete masculinization and delayed puberty onset in the Hom male mice. Despite these differences, testicular histology in adult animals did not reveal obvious differences in seminiferous tubule diameter among the Hom males and their WT and Het littermates (Fig. 1C–F, $P = 0.742$ for WT vs. Het, $P = 0.8977$ for WT vs. Hom, and $P = 0.4946$ for Het vs. Hom, based on one-way ANOVA and Tukey's post hoc test). Similarly, there were no significant differences in the perimeter (boundary) and enclosed area of seminiferous tubules among the three groups (Appendix Fig. S2A,B), suggesting that the homozygous LhrT465I mutation does not affect the viability and structural development or organization of gonadal cells in male mice.

## Impaired calcium signaling responses to LH in Leydig cells of homozygous LhrT465I males

The LHR is primarily G$_s$- and G$_q$-coupled (Breen et al, 2013; Derkach et al, 2017; Egbert et al, 2019; Rajagopalan-Gupta et al, 1998). Although cAMP has been measured classically for LHR signaling (Rajagopalan-Gupta et al, 1998), more recent studies (Bahena-Alvarez et al, 2019) have used calcium imaging. While G$_s$ is classically known to increase cAMP levels (Neves et al, 2002), a recent study shows that it also increases intracellular calcium (Brands et al, 2024). Since G$_q$ itself is classically known to increase intracellular calcium levels, this makes calcium imaging the most robust approach to measure LHR signaling-related activity. Importantly, G$_q$-specific LHR signaling has been shown to be crucial for triggering ovulation (Breen et al, 2013). Therefore, to compare LH signaling in the testes of homozygous mutant males to those of control littermates, we measured calcium mobilization as a sensitive surrogate output of LHR signaling competence (Bahena-Alvarez et al, 2019). To achieve this, LhrT465I-IC mice were crossed with eROSA26-GCaMP3 mice (Paukert et al, 2014), resulting in LhrT465I-IC$^{+/-}$/eR26-GCaMP3$^{+/-}$ (control Het) and LhrT465I-IC$^{+/+}$/eR26-GCaMP3$^{+/-}$ (Hom) animals (Appendix Fig. S3A). Expression of the genetically encoded calcium biosensor GCaMP3 in the LHR cells facilitated visualization of calcium

responses to LH in live tissue, providing functional data. Calcium imaging was performed on acutely prepared slices from the testes of control Het and Hom mice. A heatmap assembled from the calcium responses of Leydig cells pooled from control Het mice showed a clear and specific increase in fluorescence intensity upon LH application, which decreased to baseline after washing out LH (Fig. 2A). Leydig cells from Hom mice showed much diminished responses to LH application (Fig. 2B). For control Het mice, the total change in fluorescence intensities during LH application was approximately five times more than the "basal" (before LH application) state (Fig. 2C; ****$P < 0.0001$ for basal [Het] vs. LH [Het], two-way ANOVA; $n = 278$ cells pooled from 3 Het mice). However, for Hom mice, the total change in fluorescence intensities during LH application was only 2.3 times more than the basal state, less than half that seen for control Het mice (Fig. 2C; ****$P < 0.0001$ for basal [Hom] vs. LH [Hom], two-way ANOVA; $n = 175$ cells pooled from 3 Hom mice; ****$P < 0.0001$ for LH [Het] vs. LH [Hom], two-way ANOVA). No significant difference in basal fluorescence intensities was observed between these groups ($P = 0.5435$ for basal [Het] vs. basal [Hom]). A similar pattern was observed for areas under the curve (AUC) of the calcium responses, with Leydig cells from Hom mice showing a smaller increase in AUC upon LH application than cells from control Het mice (Appendix Fig. S3B). Representative traces of Leydig cells from both groups demonstrate the temporal profile of responses to LH application (Fig. 2D,E), which were observed to be more heterogeneous in Leydig cells from Hom mice (Fig. 2E). Together, these results demonstrate compromised LHR signaling in Leydig cells from Hom mice.

## AI reveals inherent heterogeneity in the calcium responses of Leydig cells from heterozygous control as well as homozygous mutant males

Ex vivo calcium imaging of tissue slices yields complex calcium signaling patterns that reflect the basal/spontaneous state of the cell as well as its ability to respond to a specific ligand (LH, in the present case). This is coupled with the inherent heterogeneity (e.g., in gene and receptor expression profiles) among and within the LHR-expressing cell types and subtypes. Further, the age, hormonal, biophysical, and physiological states of the animal affect cellular signaling patterns. These culminate into immense complexity and variability in calcium responses of the cells. While

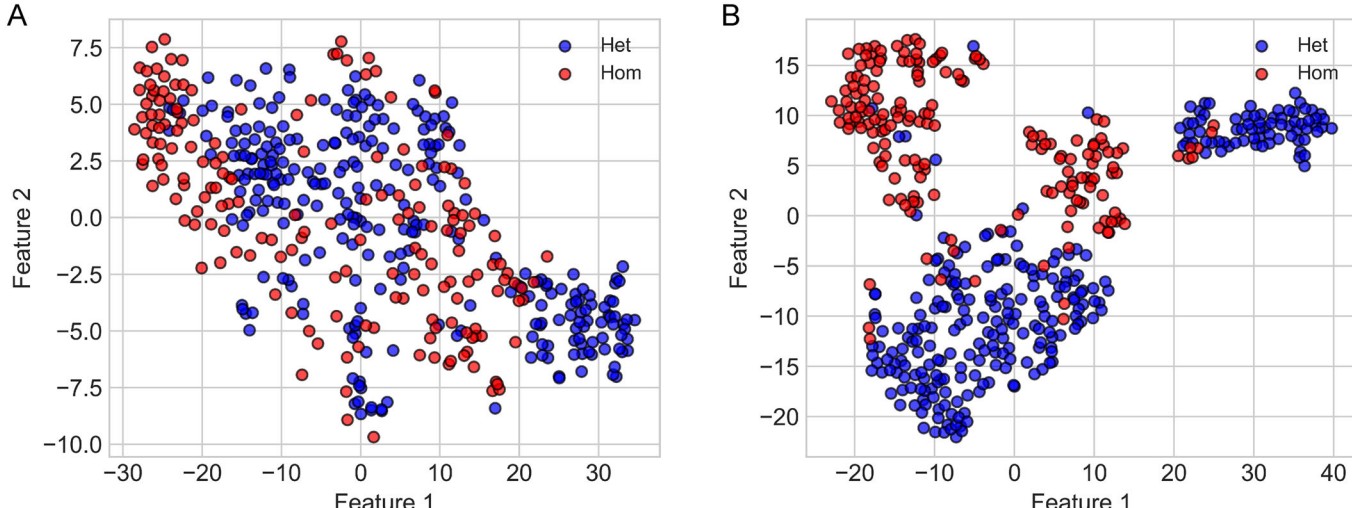

Figure 3. Artificial intelligence reveals inherent heterogeneity in the calcium responses of Leydig cells to LH.

(A) t-SNE projection of male control Het and Hom cells, mapped to 2D while preserving local structure—similar cells remain close together, approximating relative distances locally. While some overlap is observed, distinct subclusters emerge, indicating that Hom cells exhibit separable features from Het controls, though with some shared characteristics. (B) Neural network (AI model) feature space projection of male control Het and Hom cells. The network, trained to distinguish between these groups, reveals well-separated clusters, indicating distinct feature representations for control and mutant cells. The clear separation suggests strong biological differences between the two groups, as captured by the model.

traditional statistical comparisons reveal population-level differences in calcium signaling across genotypes and treatments, they do not capture the full complexity of high-dimensional single-cell response patterns. We reasoned that AI, based on a machine-learning neural network, may capture the intrinsic complexity and heterogeneity of the cells. In preliminary analyses, mean traces (solid lines) and variability (±standard deviation, shaded region) in the changes in fluorescence intensities for the Het and Hom groups showed group-level dynamics in their respective calcium responses (Appendix Fig. S4A,B, respectively). Het males showed a strong mean response with considerable variability across cells (Appendix Fig. S4A). Hom males, while still responsive, exhibited a slightly lower mean response and reduced variance, indicating that the T465I mutation dampens, but does not abolish, LHR-mediated calcium signaling in Leydig cells (Appendix Fig. S4B). The difference in peak value of the two mean signals is 0.17 (peak Het, 0.40; peak Hom, 0.23) (Appendix Fig. S4C). This group-level shift in peak activity implies a partial functional impairment that affects the collective responsiveness of Leydig cells, without completely silencing the LH-triggered signaling.

We then visualized the *t*-distributed stochastic neighbor embedding (t-SNE) (Van der Maaten and Hinton, 2008), a nonlinear dimensionality reduction technique widely used for visualizing high-dimensional data in 2D, which allows readily interpretable visualization of the biological heterogeneity of the local subclusters of cells. We selected t-SNE to represent the high-dimensional data because it best preserves local structure to reveal clusters, particularly for biological data (Kobak and Berens, 2019; Van der Maaten and Hinton, 2008). Furthermore, similar techniques are used to reveal relationships between individual cells' profiles, determining which cells behave similarly within complex, heterogeneous populations (Amir et al, 2013). Each cell's calcium trace was treated as a high-dimensional vector, with each

time-point representing one dimension (hence, 1500 dimensions in total). The t-SNE algorithm projects this data into a 2D space while preserving local relationships between cells, such that nearby points in the original space remain close in the embedding. We also applied K-means clustering to the t-SNE-embedded data (Appendix Fig. S5A–E; Appendix Table S1 [showing proportion of cells in each cluster that belong to the indicated experimental group]). We also performed an unsupervised similar K-means analysis full-dimensional clustering for this dataset (Appendix Table S2); however, this did not reveal meaningful patterns, with most of the cells (from both groups) clustering together even at K = 5, emphasizing the utility of applying t-SNE in the present context. The aim here was to determine whether the pooled calcium responses of cells from all the animals (Het as well as Hom) could be naturally grouped into distinct signaling patterns that align with their respective biological identities. The algorithm indeed identified subpopulations that were more homogeneous in their biological composition. While some overlap was observed between the cells from control Het and Hom males, a pattern of segregation also emerged (Fig. 3A). Cells from Het males formed a relatively diffuse distribution, consistent with the high variance observed in the mean trace (Appendix Fig. S4A). Hom cells appeared in specific regions of the t-SNE space, indicating reduced heterogeneity and suggesting a shift in the overall structure of LH responses in cells harboring the mutant receptors (Fig. 3A). For comparison, we have also added PCA-based plots which can show patterns in the data in the direction of highest variance and also UMAP plots to obtain the global structure of the data, as this approach best preserves the true global geometry (Becht et al, 2019) (Fig. EV3A–D). A PCA plot of the unsupervised clustering of the calcium profiles of Het and Hom Leydig cells failed to show clear patterns, with only slight differences in the distribution of Het and Hom cells being observed, without distinct clusters (Fig. EV3A). A UMAP plot of

the data revealed a more distinct distribution of the two groups, albeit with considerable overlap and very few small subclusters with only Het or Hom cells (Fig. EV3B). Supervised clustering (using AI model), on the other hand, achieved clear segregation of the Het and Hom cells, with minimal overlap, as visualized using PCA (Fig. EV3C) or UMAP (Fig. EV3D). The partial overlap between groups suggests that Hom cells still retain calcium response features reminiscent of the controls, in line with the less severe impairment observed in male Hom mutants in comparison to females (see below).

While unsupervised methods revealed structure aligned with biological conditions, they do not directly test the model's ability to predict those conditions. We used traditional machine-learning techniques, including support vector machines, Decision Trees, and Random Forest, to classify each Leydig cell as control (Het) or mutant (Hom) based on its calcium trace profile. While most of the classifiers performed well, with the top 3 performers, SVM [RBF kernel], Random Forest, and multi-layer perceptron, yielding accuracies of 78, 79, and 79%, respectively (Table 1), these approaches do not allow accurate and faithful visualization of the relative differences among the cells, which is crucial for visual interpretation of such complex biological data. Moreover, these approaches are not as efficient as neural networks in nonlinear adaptive feature-learning and scalability, which would be of paramount importance for the practical application of a potential preclinical classifier. Thus, we trained a feature-learning feedforward neural network in a supervised framework, optimized for analyzing time-series data, to classify calcium trace profiles as either control (Het) or mutant (Hom). We then projected the entire data into the feature space of the network, which, using explainable AI (XAI), captures abstract, nonlinear combinations of the original calcium signal that are most informative for distinguishing between the two groups. Biologically, it represents a learned embedding of cellular responses where functional differences in signaling dynamics, such as amplitude, wavelength, or shape, are distilled

into a lower-dimensional form that separates distinct phenotypes. Using the above framework, we trained the neural network classifier on male data with a classification accuracy of 86% on the test dataset, indicating that control Het and Hom cells exhibit clearly separable calcium dynamics. On projecting all cells into the learned feature space, the control Het and Hom cells formed distinct, well-separated clusters (Fig. 3B). This confirms that the model successfully captured key discriminative features underlying genotype-specific responses. To assess whether the supervised feature space learned by the neural network preserves biologically meaningful structure, we applied K-means clustering to the hidden-layer representations of control Het and mutant Hom Leydig cells (Appendix Fig. S5F–J; Appendix Table S3). Thus, the neural network captured latent structure in the calcium response dynamics that aligned with the biological groups, even when labels were removed. Supervised full-dimensional clustering of Het and Hom cells revealed one large cluster per group (at K > 2), with >90% purity, accurately capturing the discriminative features in calcium profiles of these cells (Appendix Table S4). However, further subclustering was conspicuously absent, compared to the results obtained using t-SNE (Appendix Table S3). Taken together, these data demonstrate the segregation of the Het and Hom Leydig cells in terms of their calcium signals, despite partial overlap between the two populations. Moreover, the AI algorithm reveals considerable inherent heterogeneity in the calcium signatures of Hom as well as Het cells, which potentially contributes to the overlap.

## Impaired estrous cycles and lack of *corpora lutea* in adult homozygous mutant females

To test the effect of the T465I Lhr mutation on puberty onset in LhrT465I-IC$^{+/+}$ females, body weight, vaginal opening, and age at first estrus were determined (Mayer and Boehm, 2011). In contrast to males, there were no significant differences in the body weights of Het and Hom mutant females when compared to WT at any of the experimental time-points (Fig. 4A; PND 24: WT vs. Hom ns $P = 0.8474$, WT vs. Het ns $P = 0.9372$, Het vs. Hom ns $P = 0.9966$; PND 42: WT vs. Hom ns $P = 0.9162$, WT vs. Het ns $P = 0.9205$, Het vs. Hom ns $P = 0.9976$; two-way ANOVA, Tukey's post hoc test). The age at vaginal opening was also not significantly different among the groups (Fig. 4B; WT vs. Hom, $P = 0.9825$; WT vs. Het, $P = 0.5242$; Het vs. Hom, $P = 0.5931$; one-way ANOVA, Tukey's post hoc test). Furthermore, the age at which the mice first exhibited cornified epithelial vaginal cytology (first estrus) did not differ among the groups (Fig. 4C; WT vs. Hom, $P = 0.374$; WT vs. Het, $P = 0.4144$; Het vs. Hom, $P = 0.0726$; one-way ANOVA, Tukey's post hoc test). These findings indicate that the T465I mutation does not alter puberty onset in female mice.

We next questioned whether adult Hom females are phenotypically distinguishable from their WT and Het littermates in terms of fertility. Altered or impaired estrous cycling is known to be a reliable readout of reproductive competence (Babwah, 2015; Mayer and Boehm, 2011; Novaira et al, 2014; Sullivan and Moenter, 2004). To monitor estrous cycles, we collected vaginal smear data for each female. The Hom mice clearly showed an absence of regular estrous cyclicity. Proestrus was not detected for these animals, and extended periods of estrus or metestrus were observed (Fig. 5C). Conversely, WT and Het mice were found to exhibit regular estrous

**Table 1. Comparison of performances of machine-learning techniques and feature-learning AI when classifying Leydig cells as Het or Hom.**

| Mathematical models | Hyperparameters | Control (Het) vs. mutant (Hom) | |
|---|---|---|---|
| | | Accuracy (%) | F1 score |
| SVM (RBF kernel) | $C = 1.0$, $\gamma = \frac{1}{n_{features} \cdot Var(Data)}$ | 78.02 | 0.77 |
| SVM (linear kernel) | $C = 1.0$, max$\_iter = 10,000$ | 71.43 | 0.7 |
| Logistic regression | $C = 1.0$, max$\_iter = 5000$ | 70.33 | 0.69 |
| Decision tree | default (gini, no depth limit) | 69.23 | 0.69 |
| K-nearest neighbors | $n_{neighbors} = 5$ | 67.03 | 0.58 |
| Gaussian Naïve Bayes | $priors = None$, $smoothing = 1 \times 10^{-9}$ | 53.85 | 0.49 |
| Random forest | $n_{estimators} = 50$ | 79.12 | 0.78 |
| Multi-layer perceptron | 1 hidden_layer_sizes = (400), relu+adam | 79.12 | 0.79 |
| Feature-learning AI | N/A | 81.32 | 0.81 |

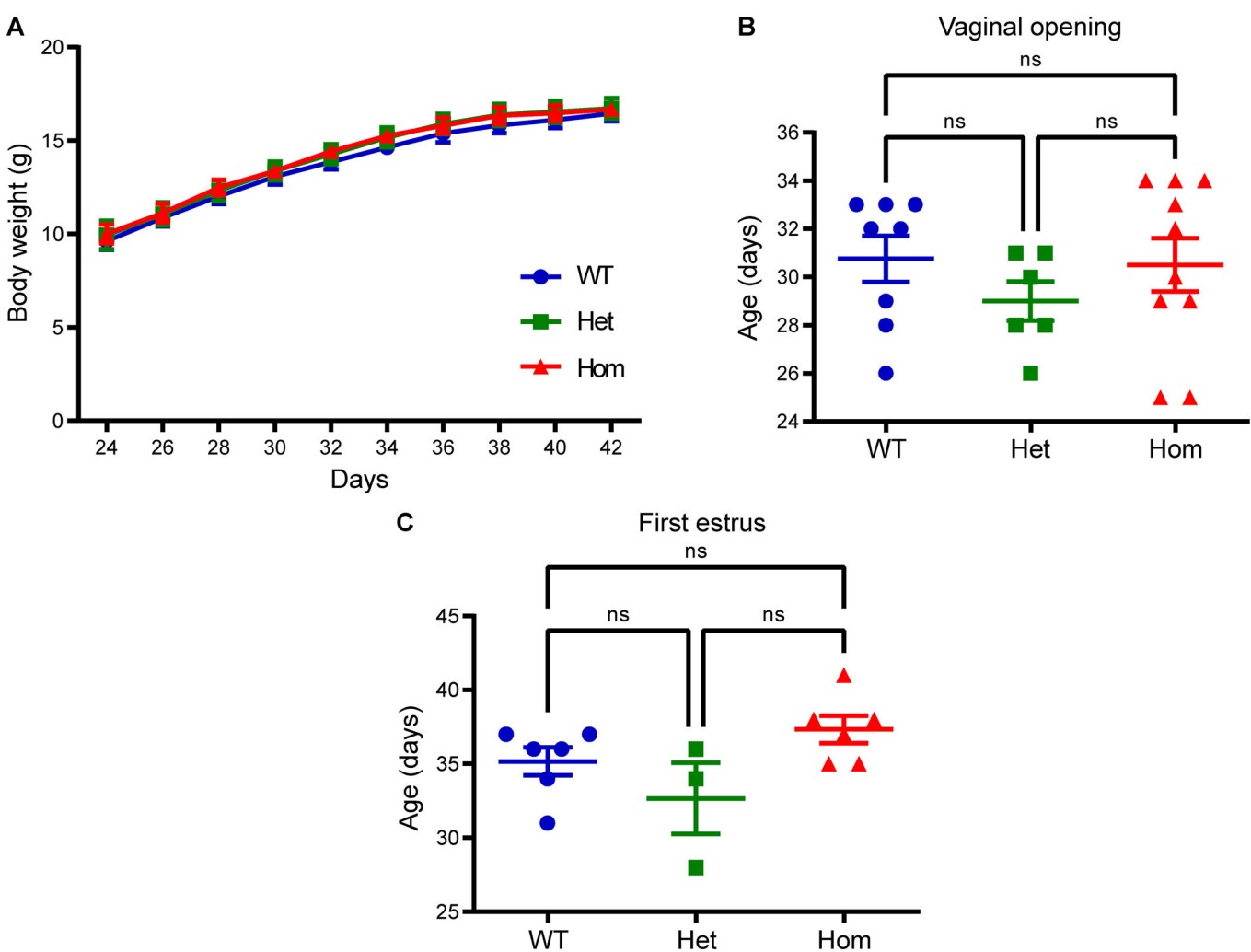

**Figure 4. The LhrT465I mutation does not affect body weight and puberty onset in females.**

(A) Female homozygous mutant mice (LhrT465I-IC$^{+/+}$, Hom) and their heterozygous (LhrT465I-IC$^{+/-}$, Het) and wild-type (LhrT465I-IC$^{-/-}$, WT) littermates were weighed every other day from PND 24 to PND 42. The body weights of the Hom mice were not different from those of their Het and WT littermates throughout the entire study period. $N = 9$ (WT), 6 (Het), and 12 (Hom) mice. WT vs. Hom at PND 24, $P = 0.8474$; PND 26, $P = 0.9235$; PND 28, $P = 0.7178$; PND 30, $P = 0.8551$; PND 32, $P = 0.5534$; PND 34, $P = 0.4961$; PND 36, $P = 0.79$; PND 38, $P = 0.6862$; PND 40, $P = 0.8118$; and PND 42, $P = 0.9162$. WT vs. Het at PND 24, $P = 0.9372$; PND 26, $P = 0.96$; PND 28, $P = 0.9124$; PND 30, $P = 0.8891$; PND 32, $P = 0.8160$; PND 34, $P = 0.7053$; PND 36, $P = 0.7798$; PND 38, $P = 0.744$; PND 40, $P = 0.829$; and PND 42, $P = 0.9205$. Het vs. Hom at PND 24, $P = 0.9966$; PND 26, $P = 0.9977$; PND 28, $P = 0.9663$; PND 30, $P > 0.9999$; PND 32, $P = 0.9714$; PND 34, $P = 0.992$; PND 36, $P = 0.9934$; PND 38, $P = 0.9992$; PND 40, $P = 0.9969$; and PND 42, $P = 0.9976$. (B) Age of the mice at vaginal opening was not significantly different among the groups. $N = 8$ (WT), 6 (Het), and 10 (Hom) mice. WT vs. Hom, $P = 0.9825$; WT vs. Het, $P = 0.5242$; Het vs. Hom, $P = 0.5931$. (C) After vaginal opening, the age of the mice at which first estrus was detected (based on vaginal cytology) was not significantly different among the groups. $N = 6$ (WT), 3 (Het), and 6 (Hom) mice. WT vs. Hom, $P = 0.374$; WT vs. Het, $P = 0.4144$; Het vs. Hom, $P = 0.0726$. Data information: In (A–C), data were presented as mean ± SEM. In (A), *$P \leq 0.05$ (two-way ANOVA with Tukey's post hoc test). In (B, C), *$P \leq 0.05$ (one-way ANOVA with Tukey's post hoc test). Source data are available online for this figure.

cycles, taking into consideration the natural variation across animals and even between two cycles of the same animal (Fig. 5A,B).

The inability of adult Hom females to reach the proestrus stage of the estrous cycle indicates a likely effect of the T465I mutation on ovulation. Ovarian histology revealed developing and mature follicles as well as *corpora lutea* in the ovaries of adult WT and Het females (Fig. 5D,E). However, although the ovaries of the adult Hom females contained developing and mature follicles, they also contained several large "empty" follicles (Fig. 5F) (Chen et al, 2021; Williams and Stanley, 2011). Moreover, the ovaries were completely

devoid of *corpora lutea* (Fig. 5F), a structure formed after oocyte release. Thus, the T465I mutation of LHR affects ovarian cycles in female mice, with the lack of *corpora lutea* and presence of empty follicles indicating absence of ovulation.

## Absent calcium signaling responses to LH in ovarian cells of homozygous LhrT465I females

LH-induced calcium signaling has previously been reported in ovarian cells (Bahena-Alvarez et al, 2019). To compare LH signaling in the ovary of Hom females to control littermates, we

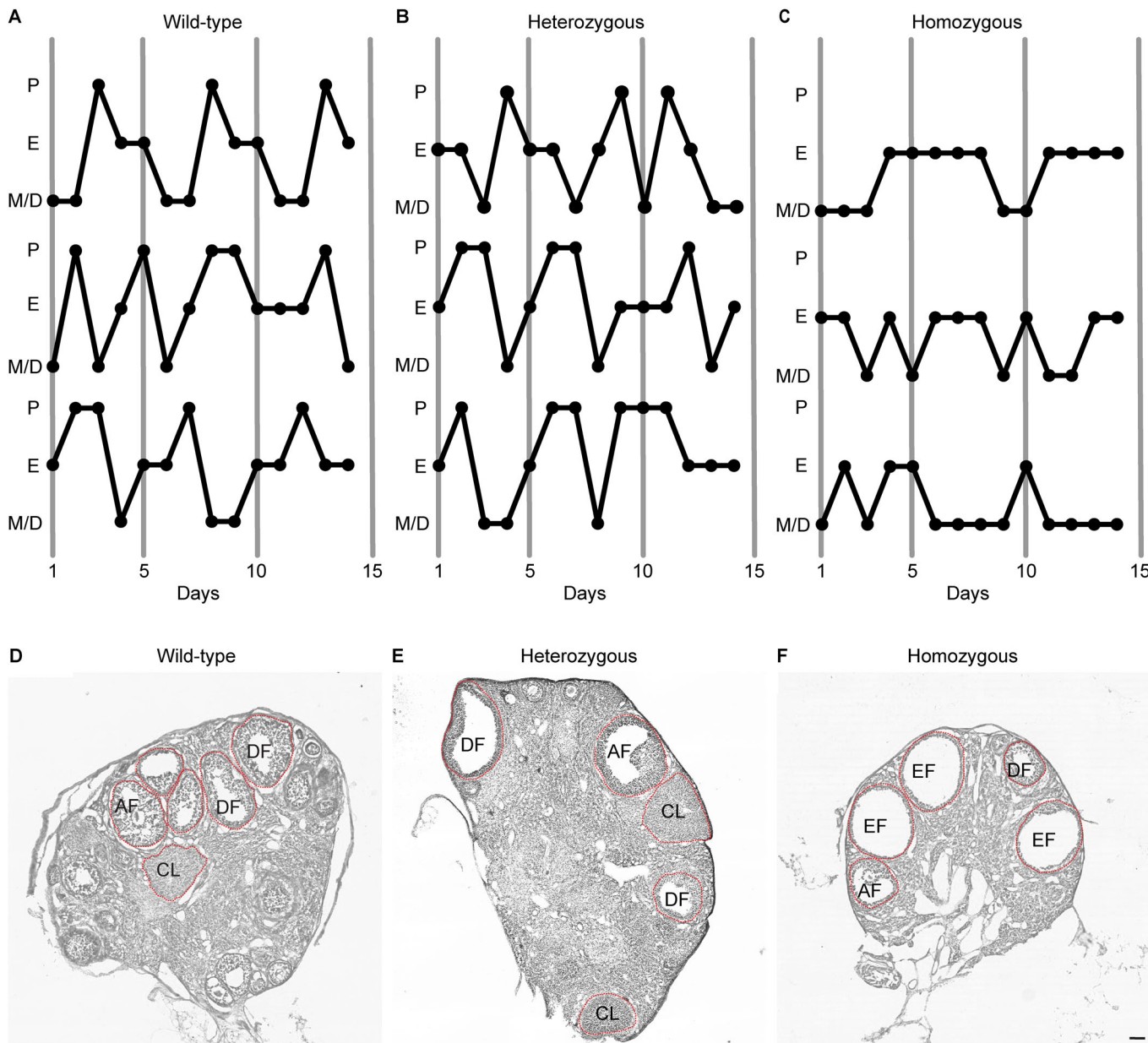

**Figure 5. Adult homozygous mutant females (LhrT465I-IC⁺/⁺) exhibit irregular estrous cycles and their ovaries do not contain *corpora lutea*.**

(A–C) Vaginal cytology across 2 weeks reveals lack of regular estrous cycling of homozygous mutant female mice (**C**) but not their WT (LhrT465I-IC⁻/⁻) (**A**) and heterozygous (LhrT465I-IC⁺/⁻) (**B**) littermates. P proestrus, E estrus, M/D metestrus/diestrus. (D–F) Hematoxylin-eosin staining of ovaries from wild-type (**D**) and heterozygous (**E**) mice show developing and mature follicles as well as *corpus luteum*. However, ovaries from homozygous mutant females (**F**) show presence of several "empty" follicles, but no *corpora lutea*. AF antral follicle, DF developing follicle, EF empty follicle, CL *corpus luteum*. Scale bar: 100 μm. Source data are available online for this figure.

prepared acute slices from the ovaries of control LhrT465I-IC⁺/⁻/ eR26-GCaMP3⁺/⁻ (control Het) and LhrT465I-IC⁺/⁺/eR26-GCaMP3⁺/⁻ (Hom) mice. LH application triggered increased intracellular calcium levels in ovarian cells from control Het mice, but not in those from Hom mice (Fig. 6A–C). A heatmap of calcium responses of ovarian cells pooled from control Het mice showed a clear and specific increase in fluorescence intensity upon LH application (Fig. 6A); however, no significant change in fluorescence intensity was observed upon applying LH to tissue

slices from Hom mice (Fig. 6B). The total change in fluorescence intensities before ("basal") and during application of LH was statistically significant for control Het mice, but not for Hom mice (Fig. 6C; ****$P < 0.0001$, two-way ANOVA, $n = 401$ cells pooled from 5 control Het mice; ns, $P = 0.9336$, $n = 203$ cells pooled from 5 Hom mice). Quantification of the AUC for the calcium responses showed that ovarian cells from Hom mice had no significant increase upon LH application, in contrast to the cells from the control Het mice (Appendix Fig. S3C). Representative traces of

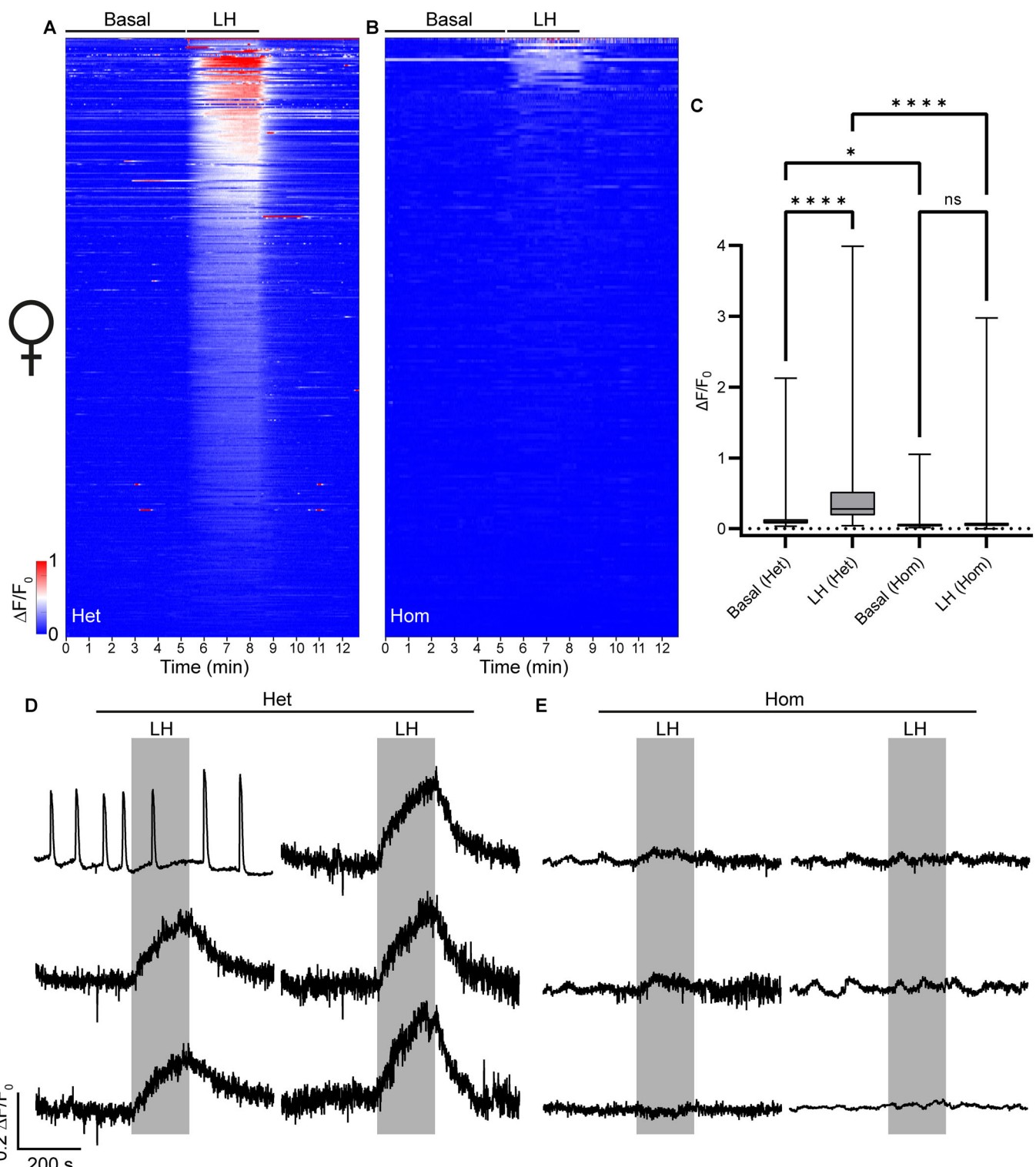

ovarian cells from control Het animals (Fig. 6D) showed that the increase in fluorescence intensities was temporally specific to the LH application, but no such increase was observed for Hom mice (Fig. 6E). The first calcium trace in Fig. 6D potentially represents a subset of LHR-expressing ovarian cells from Het mice that are distinct from the rest of the population, highlighting some apparent

heterogeneity in the calcium profiles of ovarian LHR cells. To confirm that ovarian cells from Het (LhrT465I-IC$^{+/-}$) female mice do not differ in their calcium response to cells from WT (LhrT465I-IC$^{-/-}$) female mice and are therefore comparable to WT controls, we microinjected AAV9-GCaMP3 into their ovaries. Ex vivo calcium imaging of these ovarian slices did not show any difference

◀ **Figure 6. Lhr-expressing ovarian cells of LhrT465I-IC$^{+/-}$/eR26-GCaMP3$^{+/-}$ (Het) mice respond to LH, but not those of LhrT465I-IC$^{+/+}$/eR26-GCaMP3$^{+/-}$ (Hom) mice.**

(A) Heatmap showing collected normalized fluorescence intensities from 401 ovarian cells of Het mice (pooled from 5 mice; this is the control group). The heatmap consists of three sections. The first part indicates the time from the start of the experiment up to LH application. This part of the heatmap shows the spontaneous/basal activity of the ovarian cells. The second part (marked by "LH") indicates the bath application of 1 IU/ml LH. The third part indicates washing of the LH with recording buffer. Cells in the heatmap have been represented in the descending order of fluorescence intensities of their Ca$^{2+}$ responses to LH. Scale of the changes in fluorescence intensities in the heatmap are indicated by the color-coded bar. (B) Heatmap showing collected normalized fluorescence intensities from 203 ovarian cells of Hom mice (pooled from 5 mice). (C) Box-and-whisker plot showing quantification of the total change in fluorescence intensities for all cells before ("Basal") and during LH application, for Het and Hom mice. Asterisks depict statistical significance (****$P < 0.0001$ for Basal [Het] vs. LH [Het] and LH [Het] vs. LH [Hom], and *$P = 0.0149$ for Basal [Het] vs. Basal [Hom]), ns ($P = 0.9336$ for Basal [Hom] vs. LH [Hom]) indicates lack of statistical significance. For Het, $n = 401$ cells (from 5 mice). For Hom, $n = 203$ cells (from 5 mice). (D, E) Representative traces of 6 ovarian cells each from Het (D) and Hom (E) mice, showing specific response to LH application (indicated by the gray rectangle). Data information: In (C), data were presented as a box-and-whisker plot. The boxes themselves span from the first quartile (25%) to the third quartile (75%), representing the interquartile range where the central 50% values lie. Inside each box, a line denotes the median value. The whiskers of each boxplot extend from the ends of the box to the minimum and maximum values. ****$P < 0.0001$ (two-way ANOVA with Bonferroni post hoc test). Source data are available online for this figure.

in the basal level of intracellular calcium or in the responses to LH application (Appendix Fig. S6A–C). These results demonstrate that disruption of LH receptors in ovarian cells from female Hom mice abolishes their ability to respond to acute LH application via signaling-induced calcium release.

## Homozygous mutant males are fertile, but females are infertile

We next asked whether adult homozygous LhrT465I mutant males are fertile. We set up breeding pairs each with a WT control female and either a WT control, Het control, or Hom male. Females were then checked regularly for signs of pregnancy. The Hom and Het males were fertile and bred normally, as shown by the average frequency of litters sired (Fig. EV4A). Further, the average time elapsed between litters was comparable across the groups (Fig. EV4B; $P = 0.3384$ for WT vs. Het, $P = 0.8246$ for WT vs. Hom, $P = 0.6369$ for Het vs. Hom; one-way ANOVA, $N = 3$ mice per group). There was also no statistically significant difference in the average number of pups born per litter in each group (Fig. EV4C; $P = 0.8292$ for WT vs. Het, $P = 0.7029$ for WT vs. Hom, $P = 0.9709$ for Het vs. Hom; one-way ANOVA, $N = 3$ mice per group), average survival of pups per litter (Fig. EV4D; $P = 0.5622$ for WT vs. Het, $P = 0.8719$ for WT vs. Hom, $P = 0.8404$ for Het vs. Hom; one-way ANOVA, $N = 3$ mice per group), or fertility index (a composite measure of the average rate of birth of pups as well as the litter size (Handelsman et al, 2020)) (Fig. EV4E; $P = 0.7842$ for WT vs. Het, $P = 0.9604$ for WT vs. Hom, $P = 0.6318$ for Het vs. Hom; one-way ANOVA, $N = 3$ mice per group). Thus, the T465I LHR mutation does not affect fertility in homozygous mutant male mice.

To examine whether adult homozygous LhrT465I mutant females are fertile, similar breeding trials were set up. In this case, Het or Hom females were individually bred with WT control male mice; WT females were bred with Het male mice so that the offspring would have experimentally useful genotypes, thus avoiding waste. Again, females were checked regularly for signs of pregnancy. The WT and Het females exhibited similar breeding patterns, although the Het females showed a lower average frequency of giving birth (Fig. EV4F). In sharp contrast, Hom females were found to be completely infertile, with none of the females giving birth during the 100 days of observation (Fig. EV4F). Although the mean inter-litter interval for Het females showed a trend towards higher values, it was not significantly different from

that of the WT group (Fig. EV4G; $P = 0.1638$, based on unpaired two-tailed *t*-test). The average number of pups delivered per litter was similar between WT and Het females (Fig. EV4H; $P = 0.7863$, based on unpaired two-tailed Student's *t*-test). Although the Het females displayed high intra-group variability, the survival rate of pups was not significantly different from that of the WT females (Fig. EV4I; $P = 0.3909$, based on unpaired two-tailed *t*-test). The composite fertility index for females showed a trend towards a lower value for Het females than for WT females, but this difference was not statistically significant (Fig. EV4J; $P = 0.1076$, based on unpaired two-tailed Student's *t*-test). These results demonstrate that WT and heterozygous mutant female mice have similar fertility; however, homozygous mutant female mice are infertile.

## LHR-Chap administration regularizes estrous cycling and facilitates ovulation in homozygous mutant females

As described above, LHR-Chap was identified as a low molecular weight, orally active agonist of the LHR (van de Lagemaat et al, 2009; van Koppen et al, 2008; van Straten et al, 2002). It is a thienopyri(mi)dine that also acts as a PC for the misfolded T461I mutant human LHR in heterologous cells in vitro, trafficking it to the cell membrane where it can then successfully bind, and be activated by the hormone (Newton et al, 2021). As the female homozygous mutant mice harboring this mutation demonstrated a clearly impaired reproductive phenotype, these were an ideal test group to investigate whether systemic administration of the PC LHR-Chap could restore their disrupted estrous cycles and elicit ovulation. Estrous cycles of adult Hom females were recorded for 23 days, during which none of the mice exhibited vaginal cytology corresponding to the proestrus stage. The Hom females in the control treatment group were then administered 500 IU/kg human chorionic gonadotropin (HCG, a more potent homolog of LH conventionally used to induce ovulation in assisted reproduction protocols) (van de Lagemaat et al, 2009) for 3 consecutive days, and subsequently, on every alternate day for 11 days. HCG administration neither stimulated proestrus nor restored a normal cycle (Fig. 7A). The test group of Hom mice received 25 mg/kg LHR-Chap in place of HCG using the same treatment regimen (van de Lagemaat et al, 2009). Strikingly, the mice that received LHR-Chap exhibited more regular estrous cycles, with the detection of the proestrus stage, indicating ovulation (Fig. 7B). Therefore, LHR-

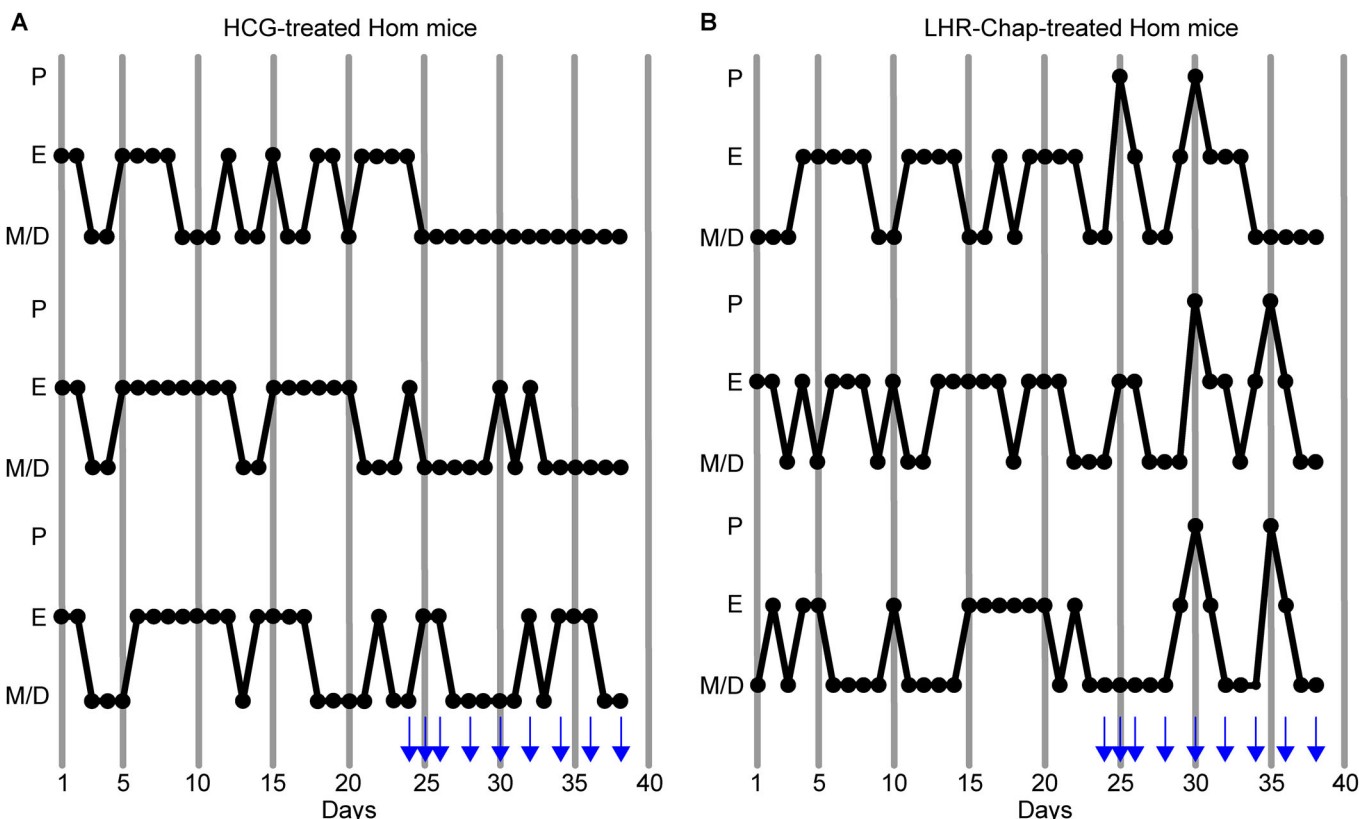

**Figure 7. LHR-Chap administration regularizes estrous cycling in LhrT465I-IC⁺/⁺ (Hom) females.**

(A, B) Estrous cycle stages were recorded for 23 days for adult Hom females. They were then i.p.-injected with 500 IU/kg HCG or 25 mg/kg LHR-Chap on the days indicated by the blue arrows until the 37th day. Administration of HCG did not alter estrous cycling in the mice (A), but LHR-Chap injection caused a relative regularization of the estrous cycles (B), notably leading to detection of proestrus stages in the injected mice. P proestrus, E estrus, M/D metestrus/diestrus.

Chap administration to adult Hom females appears to restore LHR signaling in the ovarian cells presumably by chaperoning the mutant receptors to the cell membrane where they can bind and respond to circulating LH, and possibly also by subsequently directly activating the membrane-bound receptors, as we have shown in vitro (Newton et al, 2011; Newton et al, 2021).

Following completion of the treatment regimen, ovaries were removed from the mice and processed for histological assessment. Hematoxylin-eosin staining of ovarian sections from these mice showed that HCG-injected Hom control mice exhibited the same ovarian histology described for untreated Hom mice (Fig. 5F), namely presence of large, apparently empty follicles and the absence of a *corpus luteum* (Fig. 8A). However, the ovarian sections from the LHR-Chap-injected Hom mice showed the presence of developing follicles at various stages and also the presence of multiple *corpora lutea*, an indicator of ovulation (Fig. 8B). These results indicate that systemic LHR-Chap administration of facilitates ovulation in adult homozygous mutant female mice.

## LHR-Chap administration functionally rescues LHR signaling in ovarian cells of homozygous mutant females

To analyze whether treatment of adult Hom females with LHR-Chap leads to functional rescue of LHR signaling in ovarian cells,

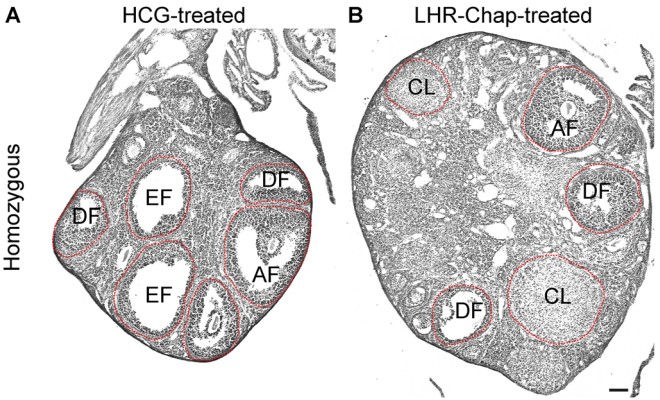

**Figure 8. LHR-Chap administration leads to formation of *corpora lutea* in ovaries of LhrT465I-IC⁺/⁺ (Hom) mice.**

(A) Systemic administration (i.p. injection) of 500 IU/kg HCG to Hom females did not lead to any detectable histological changes. Representative images of ovaries from HCG-treated Hom mice show the presence of large follicles, but no *corpus luteum*. (B) In contrast, ovaries from LHR-Chap-treated Hom mice exhibited the presence of *corpora lutea*, indicating successful ovulation. AF antral follicle, DF developing follicle, EF empty follicle, CL *corpus luteum*. Scale bar: 100 μm. Source data are available online for this figure.

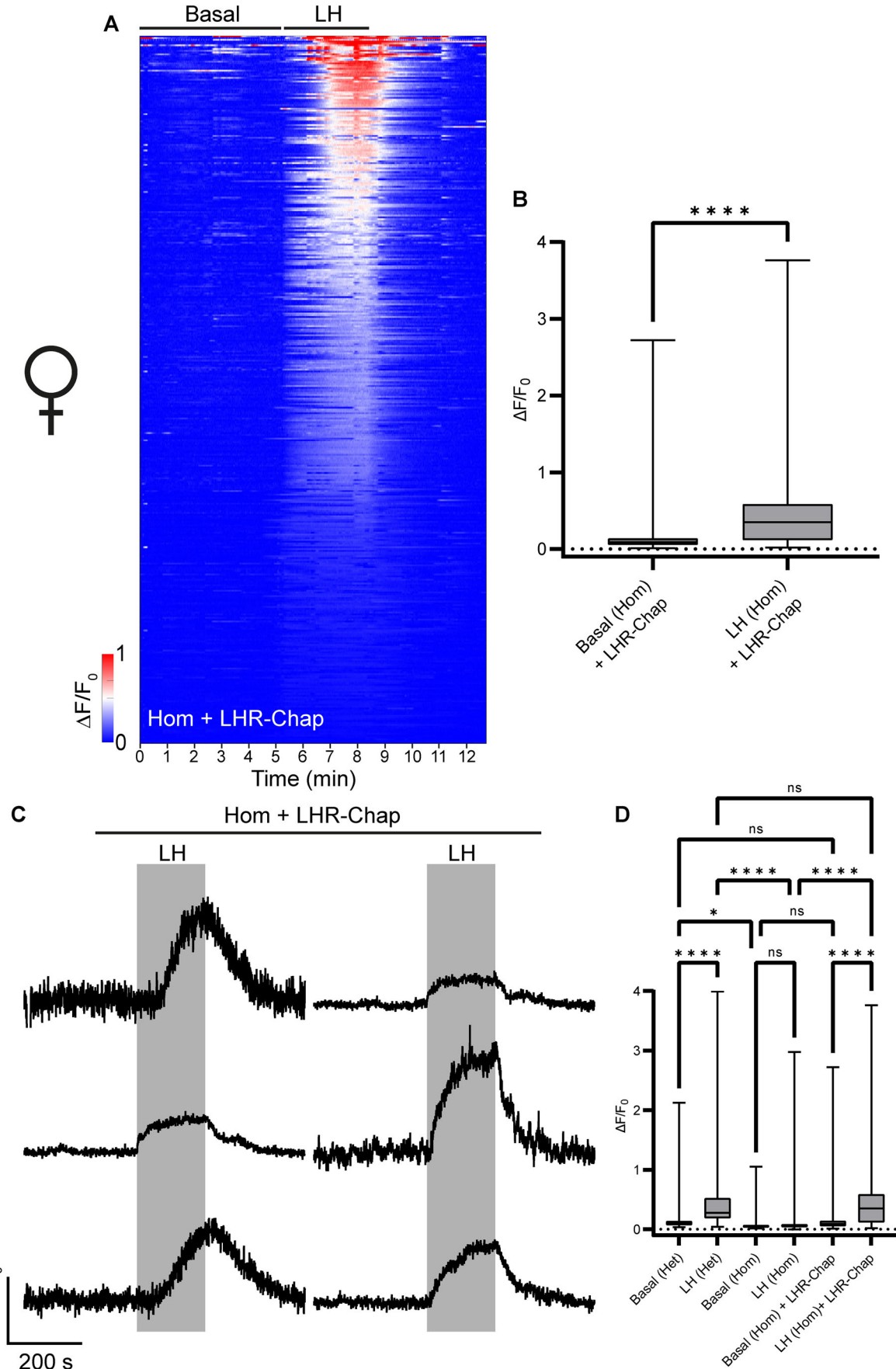

**Figure 9.   LHR-Chap administration functionally rescues LHR signaling in ovarian cells of LhrT465I-IC$^{+/+}$/eR26-GCaMP3$^{+/-}$ (Hom) mice.**

(A) Heatmap shows collected normalized fluorescence intensities from 391 ovarian cells of Hom mice (pooled from 3 mice). The heatmap comprises three sections. The first part indicates the time from the start of the experiment up to LH application. This part of the heatmap shows the spontaneous/basal activity of the ovarian cells. The second part (marked by "LH") indicates the bath application of 1 IU/ml LH. The third part indicates washing of the LH with recording buffer. Cells in the heatmap have been represented in the descending order of fluorescence intensities of their Ca$^{2+}$ responses to LH. Scale of the changes in fluorescence intensities in the heatmap are indicated by the color-coded bar. (B) Box-and-whisker plot showing quantification of the total change in fluorescence intensities for all cells before ("Basal") and during LH application, for LHR-Chap-treated Hom mice. Asterisks depict statistical significance (****$P < 0.0001$, $n = 391$ cells from 3 mice). (C) Representative traces of 6 ovarian cells from LHR-Chap-treated Hom mice, showing specific response to LH application (indicated by the gray rectangle). (D) Box-and-whisker plot comparing the total change in fluorescence intensities for all cells before and during LH application, for untreated Het ($n = 401$, $N = 5$; this is the control group), untreated Hom ($n = 203$, $N = 5$), and LHR-Chap-treated Hom mice. Asterisks depict statistical significance (****$P < 0.0001$ for Basal [Het] vs. LH [Het], LH [Het] vs. LH [Hom], and LH [Hom] vs. LH [Hom] + LHR-Chap; *$P = 0.05$ for Basal [Het] vs. Basal [Hom]), ns ($P > 0.9999$ for Basal [Hom] vs. LH [Hom], Basal [Het] vs. Basal [Hom] + LHR-Chap, and LH [Het] vs. LH [Hom] + LHR-Chap; $P = 0.1549$ for Basal [Hom] vs. Basal [Hom] + LHR-Chap) indicates lack of statistical significance. Data information: In (B, D), data were presented as box-and-whisker plots. The boxes themselves span from the first quartile (25%) to the third quartile (75%), representing the interquartile range where the central 50% values lie. Inside each box, a line denotes the median value. The whiskers of each boxplot extend from the ends of the box to the minimum and maximum values. In (B), ****$P < 0.0001$ (paired two-tailed Student's $t$-test). In (D), *$P \leq 0.05$ and ****$P < 0.0001$ (two-way ANOVA with Bonferroni post hoc test). Source data are available online for this figure.

LHR-Chap was administered to LhrT465I-IC$^{+/+}$/eR26-GCaMP3$^{+/-}$ mice following the regimen described above. After confirming the regularization of estrous cycles (Appendix Fig. S7), acute ovarian slices were prepared and LH-induced calcium responses measured. Strikingly, contrary to the ovarian cells from untreated Hom mice (Fig. 6B), LH application elicited a robust calcium response from the ovarian cells from the LHR-Chap-treated mice (Fig. 9A). The total change in fluorescence intensities for this calcium response to LH was significantly greater than the basal level (Fig. 9B; ****$P < 0.0001$, paired two-tailed Student's $t$-test; $n = 391$ cells pooled from 3 LHR-Chap-treated Hom mice). Similarly, the AUC for the calcium responses showed a significant increase upon LH application (Appendix Fig. S3D), unlike the ovarian cells from untreated Hom mice (Appendix Fig. S3C,E). Representative traces from ovarian cells of LHR-Chap-treated Hom mice show the specificity of the response to LH application (Fig. 9C). A comparison of the basal and stimulated (LH) change in fluorescence intensities for LHR-Chap-treated Hom females with those of untreated Het and untreated Hom females show that the response to LH in the treated Hom females was significantly greater than that for the untreated Hom females (Fig. 9D; ****$P < 0.0001$ for LH [Hom] vs. LH [Hom] + LHR-Chap, two-way ANOVA; $n = 391$ cells pooled from 3 LHR-Chap-treated Hom mice and $n = 203$ cells pooled from 5 untreated Hom mice) and similar to that of the untreated Het females (ns, $P > 0.9999$ for LH [Het] vs. LH [Hom] + LHR-Chap, two-way ANOVA; $n = 391$ cells from 3 LHR-Chap-treated Hom females and $n = 401$ cells from 5 Het mice). Analysis of the AUC for these groups yielded similar results, with LHR-Chap-treated Hom females showing a very similar pattern to that of the control Het females (Appendix Fig. S3E). Interestingly, during the first minute of LH application (out of a total application duration of 3 min) there appeared to be a lower average response for the treated Hom population, with significantly lower changes in fluorescence intensities (Appendix Fig. S8A). During this period, 37% of ovarian cells from LHR-Chap-treated Hom mice were inactive (<0.1 $\Delta F/F_0$) as opposed to only 16.5% of cells from untreated control Het females (Appendix Fig. S8C). However, the total AUC during the first minute of LH application was not significantly different between control Het and LHR-Chap-treated Hom females (Appendix Fig. S8B). When reviewing the last 2 min of the LH response, both changes in fluorescence intensity and AUC were not significantly different between control Het and

LHR-Chap-treated Hom females (Appendix Fig. S8A,B). These data demonstrate that administering LHR-Chap to adult Hom female mice rescues ovarian cell LH-induced calcium signals, although with delayed kinetics when compared to control Het mice.

## Ovarian cells from Hom mice respond heterogeneously to LHR-Chap treatment, as evidenced by AI-assisted data analysis

Calcium traces of the changes in fluorescence intensities showed robust response of the control Het ovarian cells to LH stimulation, with a mean signal peak of 0.28 (Appendix Fig. S9A). In stark contrast, ovarian cells from Hom mutant females show an almost flat response, with a drastically reduced peak of only 0.055 (Appendix Fig. S9B). LHR-Chap treatment seemingly restored the response in Hom females, resulting in a mean signal peak of 0.32 (Appendix Fig. S9C). Although variance remained elevated, the temporal dynamics and overall response profile of the treated group closely resembled the control signal pattern (Appendix Fig. S8D).

We next applied t-SNE (unsupervised clustering) to the time-series data of calcium signals from individual ovarian cells across all three experimental groups (Fig. 10A). Cells from Hom mutant females (red) were distributed predominantly in a dense region on the left side of the plot, indicating a restricted and homogeneous signaling profile with limited variability, consistent with the flat response observed in time-series traces (Appendix Fig. S9B). In contrast, control Het cells (blue) were broadly distributed across the embedding space, reflecting greater heterogeneity and the presence of more complex LH-response dynamics (Fig. 10A). A substantial portion of the LHR-Chap-treated Hom cells (green) overlapped with the Het cluster, suggesting that chaperone-mediated rescue restores signaling patterns that are similar to those of control Het cells. At the same time, a fraction of the treated cells retained proximity to the Hom cluster, revealing partial rescue or heterogeneity in treatment efficacy at the single-cell level.

To assess whether LH response phenotypes can be systematically grouped, we applied unsupervised K-means clustering to the t-SNE-embedded ovarian cell data from Het, Hom, and LHR-Chap-treated Hom groups (Appendix Fig. S10A–E; Appendix Table S5). Patterns emerging upon varying the K from 2 to 5 suggest that the chaperone-based rescue shifts the calcium signaling of Hom ovarian cells closer to that of the control Het cells, while

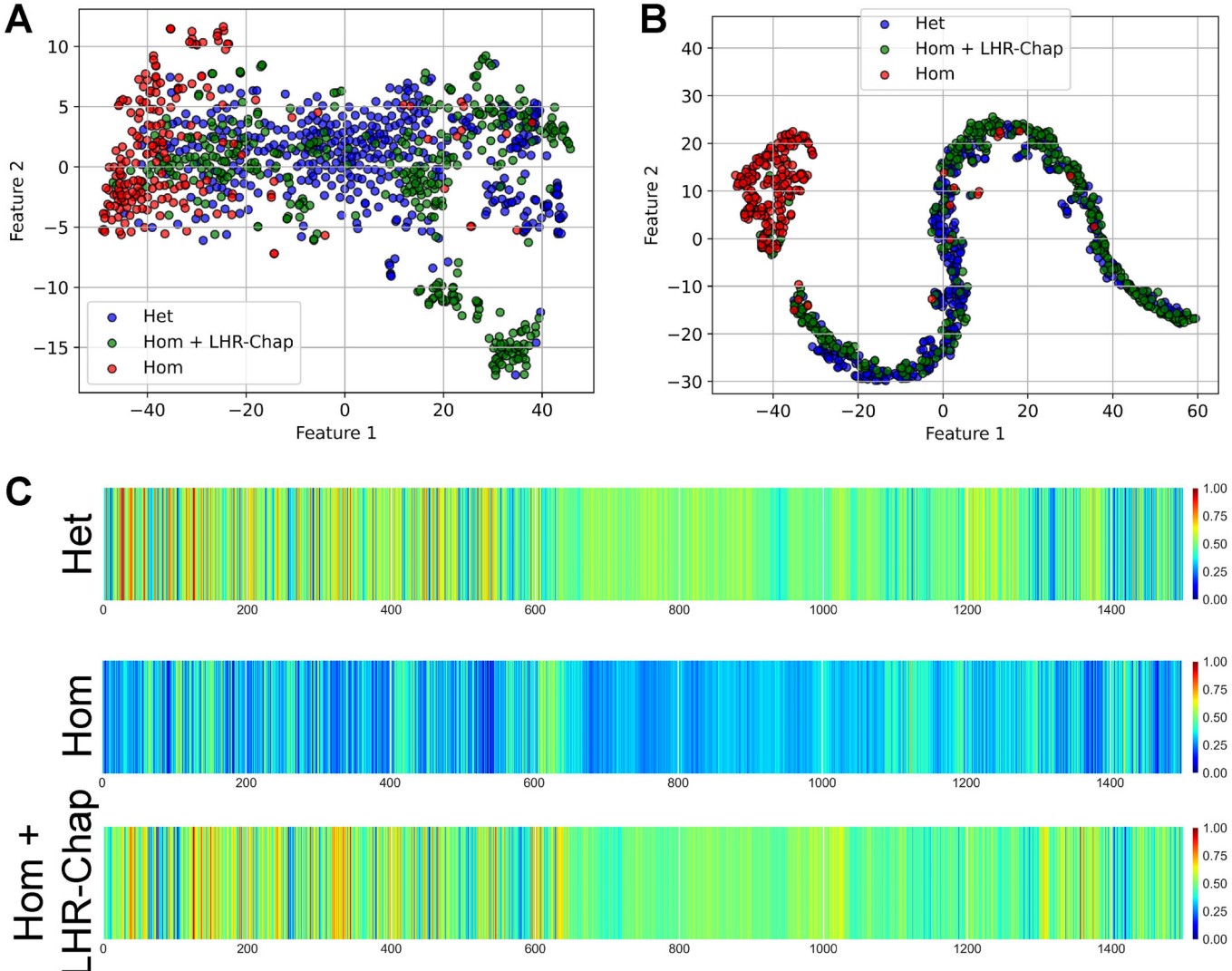

**Figure 10. Trained neural network identifies unique features to distinguish homozygous mutant ovarian cells based on calcium signals, even before LH application.**

(A) t-SNE projection of female cells, reducing high-dimensional data to 2D while preserving local structure: similar cells remain close together, approximating relative distances locally. Control Het (blue) cells form a distinct cluster, while Hom mutants (red) separate from controls. LHR-Chap-treated mutants (green) partially overlap with controls, suggesting a shift towards a control-like profile. Projection of control Het (blue), Hom mutant (red), and LHR-Chap-treated Hom mutant (green) cells is shown. (B) AI-enabled projection of female cells onto the model's feature space accurately and faithfully depicts relative distances globally, demonstrating the true positions of the cells based on their calcium signature. (C) Grad-CAM (Gradient-weighted Class Activation Mapping) visualization of feature activations in the neural network trained on female Het and Hom cells. Grad-CAM highlights the most important features the model uses for classification, providing insight into its decision-making process. Control Het (top) and LHR-Chap-treated Hom (bottom) cells activate similar features, while untreated Hom mutant cells (middle) show distinct activation patterns. This indicates that the model identifies different biological characteristics in Hom mutants, reinforcing their separation from the other groups in feature space. Importantly, this reveals that the model heavily relies on calcium signatures before and at the time of LH application to make its decision, indicating inherent differences in the groups that are predictive of their calcium response to LH.

still maintaining some of the distinct features of the untreated Hom, possibly hinting at a specific aspect of the calcium signaling to be impacted more by the treatment than the other. Unsupervised full-dimensional clustering of the cells also achieved partial segregation of these cells, with some large unresolved clusters remaining (Appendix Table S6).

PCA plot of unsupervised clustering of the calcium profiles of control Het, untreated mutant Hom, and LHR-Chap-treated mutant ovarian cells did not reveal distinct clusters, failing to resolve the different groups meaningfully, although the Hom cells clearly crowded together at one side of the plot (Fig. EV5A). UMAP revealed better clustering of the data (Fig. EV5B), with the overall distribution being somewhat similar to that achieved by t-SNE (Fig. 10A). PCA plot of supervised clustering (AI model) of these cells did not offer meaningful separation of the experimental groups (Fig. EV5C). However, UMAP representation of supervised clustering of the cells revealed a tight cluster of untreated Hom cells while the control Het and LHR-Chap-treated Hom cells clustered closer to each other, showing some subclusters that were enriched in either Het or LHR-Chap-treated Hom cells (Fig. EV5D).

**Table 2. Comparison of performances of machine-learning techniques and feature-learning AI when classifying ovarian cells as Het or Hom.**

| Mathematical models | Hyperparameters | Control (Het) vs. mutant (Hom) | | LHR-Chap-treated cells classified as control (Het) (%) |
| --- | --- | --- | --- | --- |
| | | Accuracy (%) | F1 score | |
| SVM (RBF kernel) | $C = 1.0$, $\gamma = \frac{1}{n_{features} \cdot Var(Data)}$ | 90.91 | 0.91 | 76.98 |
| SVM (linear kernel) | $C = 1.0$, $\max\_iter = 10,000$ | 76.33 | 0.8 | 66.75 |
| Logistic regression | $C = 1.0$, $\max\_iter = 5000$ | 85.12 | 0.85 | 74.68 |
| Decision tree | default (gini, no depth limit) | 81.82 | 0.82 | 80.31 |
| K-nearest neighbors | $n_{neighbors} = 5$ | 78.51 | 0.79 | 62.4 |
| Gaussian Naïve Bayes | $priors = None$, $smoothing = 1 \times 10^{-9}$ | 68.6 | 0.69 | 61.38 |
| Random Forest | $n_{estimators} = 50$ | 93.39 | 0.93 | 83.63 |
| Multi-layer perceptron | 1 hidden_layer_sizes = (400), relu+adam | 90.91 | 0.91 | 73.91 |
| Feature-learning AI | N/A | 94.21 | 0.94 | 86.96 |

Standard machine-learning techniques classified the cells into their respective experimental groups with remarkable accuracy, demonstrating the robustness of the biological data (top 3 performers, again SVM [RBF kernel], Random Forest, and multi-layer perceptron, yielding 90.91, 93.39, and 94.21% accuracy; our AI-based neural network showed 94.21% accuracy; Table 2). Visualization of the analysis performed on ovarian cells by the trained neural network revealed that the LHR-Chap-treated Hom cells clustered with the control Het cells, and were largely separated from the untreated Hom cells (Fig. 10B). This result is particularly compelling because the network was not trained on LHR-Chap-treated Hom data, yet it mapped these cells into the control Het region of the feature space. Applying K-means clustering to t-SNE of the feature representations learned for segregating control Het cells from Hom cells revealed that untreated mutant Hom cells group separately from both Het and LHR-Chap-treated Hom cells (Appendix Fig. S10F–J; Appendix Table S7). Notably, the treated cells reliably clustered with controls rather than with untreated mutants. We also found that this AI model did not distinguish between ovarian cells from AAV-GCaMP3-microinjected WT and Het mice (Appendix Fig. S11A–F), indicating that ovarian cells from the WT mice behave similarly to those from Het mice (in terms of calcium profiles). Supervised full-dimensional clustering of this dataset also demonstrated the robust LHR-Chap-mediated rescue of calcium signaling, including LH response, to a control-like state (Appendix Table S8). These results corroborate our earlier analyses and underscore that the neural network has distilled meaningful discriminative features from calcium trace profiles that reflect underlying biological differences.

Gradient-weighted class activation mapping (Grad-CAM) analysis (Selvaraju et al, 2017) further revealed the salient features of the calcium recording of each cell that the algorithm uses to categorize the cells into the three groups (Fig. 10C). The algorithm appeared to rely heavily on inherent calcium signatures during the pre-treatment ("basal") part of the experiment, indicating intrinsic differences in spontaneous calcium signaling in these cells that can be used to predict their responses to LH and subsequently categorize them accordingly. As expected, the time of LH application is also used by the algorithm to make its decision; however, the entire duration of LH application is not useful, the decision seemingly being made in the first few seconds of the calcium response. Based on this, we trained our AI model on only the first 600 frames (5 min) of the calcium recording, corresponding to the time period before LH application (termed "basal" this far). PCA, t-SNE, and UMAP representations of unsupervised clustering did not offer any meaningful pattern (Fig. EV6A–C, respectively), with cells from all three groups being distributed haphazardly across the feature space, although a few small clusters were detected for t-SNE as well as UMAP. However, the AI model revealed a clear pattern, with Hom cells mainly populating one side of the feature space, while most of the control Het and LHR-Chap-treated Hom cells clustered together across the remainder of the space (Fig. EV6D–F). Distinct subclusters were not detected with PCA, but t-SNE as well as UMAP revealed local subclusters that were frequently composed of either control or LHR-Chap-treated Hom cells, consistent with observations from analysis of the entire experimental dataset (Figs. 10A,B and EV5B,D). Analyses using machine-learning approaches further supported these findings (Table 3). This is intriguing as it uncovers differences in calcium signaling in these cells which are inherent to the Lhr mutation status, precluding the need to rely on LH response, and shows that these spontaneous pre-LH application signaling patterns can also be used to distinguish experimental groups from each other. The ability to interrogate differences within the signaling "baselines", which would not have been possible without AI, is promising for AI-assisted preclinical diagnosis. These findings also underscore the utility of AI tools (Appendix Fig. S12) to uncover hitherto inaccessible details regarding functional behavior of cells in response to a ligand as well as to pharmacological treatment(s). Here, we propose an automated pipeline that utilizes AI models to faithfully represent biological differences in cellular calcium signaling caused by a GPCR mutation (obtained using the B-Cos neural network, previously developed by us), both in absence and presence of its ligand (Appendix Fig. S12). Comparison of ROIs in confocal images marked by a human experimenter and those detected by the AI model demonstrate a high overlap, but with room to improve and refine the model further (Appendix Fig. S13A). Saliency maps corresponding to different points across the entire time-series of calcium imaging highlight the ability of our AI model to normalize saliency across the series, with the detection of ROIs remaining unaffected by local changes in fluorescence intensity (Appendix Fig. S13B).

**Table 3. Comparison of performances of machine-learning techniques and feature-learning AI when classifying ovarian cells as Het or Hom based on spontaneous/steady-state calcium profiles ("basal").**

| Mathematical models | Hyperparameters | Control (Het) vs. mutant (Hom) | | LHR-Chap-treated cells classified as control (Het) (%) |
|---|---|---|---|---|
| | | Accuracy (%) | F1 score | |
| SVM (RBF kernel) | $C = 1.0$, $\gamma = \frac{1}{n_{features} \cdot Var(Data)}$ | 88.43 | 0.88 | 62.15 |
| SVM (linear kernel) | $C = 1.0$, $\max\_iter = 10,000$ | 66.11 | 0.68 | 36.32 |
| Logistic regression | $C = 1.0$, $\max\_iter = 5000$ | 64.46 | 0.65 | 39.64 |
| Decision tree | default (gini, no depth limit) | 76.03 | 0.77 | 52.69 |
| K-Nearest Neighbor | $n_{neighbors} = 5$ | 44.63 | 0.39 | 8.18 |
| Gaussian Naïve Bayes | $priors = None$, $smoothing = 1 \times 10^{-9}$ | 68.6 | 0.69 | 46.29 |
| Random Forest | $n_{estimators} = 50$ | 85.12 | 0.84 | 73.15 |
| Multi-layer perceptron | 1 hidden_layer_sizes = (400), relu+adam | 81.82 | 0.82 | 42.2 |
| Feature-learning AI | N/A | 82.64 | 0.94 | 51.66 |

# Discussion

Based on the large involvement of GPCRs in various signaling processes, and resultant wide range of diseases caused by defective GPCRs, it is unsurprising that almost half of all therapeutics available today target GPCRs. As a result of this, there exists a vast library of small molecules (about 300–500 Da) produced by the pharmaceutical industry which are selected to be orally active (and thus cross cell membranes). Capitalizing on these properties, a large number of these molecules may be "repurposed" as PCs, which stabilize the misfolded protein and ferry them to the cell membrane, restoring normal function. Application of these molecules, most of which are highly specific and proven non-toxic, as PCs offers the opportunity to treat rare diseases (personalized medicine) arising from GPCR mutations for which there is no current treatment. To date, there are very few PCs targeting inactivating GPCR mutations (arginine vasopressin receptor 2, rhodopsin, GnRH receptor and melanocortin 4 receptor) (Ahmed et al, 2019; Bernier et al, 2006; Janovick et al, 2013; Rene et al, 2021) that have been studied in vivo. We reasoned that more extensive application of functional rescue of human mutant GPCRs could be achieved by generating mouse models bearing mutant GPCRs. We therefore scanned the patent and scientific literature for inactivating GPCR mutations in the reproductive cascade and for small molecule agonists targeting these receptors. The LHR was identified as a suitable candidate along with the published Organon suite of LHR-targeting small molecule agonists. We reasoned that the generation of a mouse strain bearing a selected LHR mutation would constitute a "proof-of-concept" as to what is required to rescue mutant GPCRs more generally.

Rescue of mutant GPCRs is a relatively embryonic therapeutic area with very far-reaching possibilities in view of the large number of GPCRs implicated in a wide range of pathologies. While promising in vitro results have been reported across several receptor types, in vivo proof-of-concept studies remain limited. In contrast to the few studies available on rescue of GPCR function (Ahmed et al, 2019; Bernier et al, 2006; Janovick et al, 2013; Ortega et al, 2022; Rene et al, 2021), we demonstrate a functional/calcium signaling rescue in adult females in the most sexually dimorphic physiological system in both mice and humans. Also, our study is

the only one to use an allosteric agonist which ferries the mutant receptor to the cell surface and at the same time stimulates the receptor. Other studies have used antagonists as PCs (Janovick et al, 2013), which then require complex manipulations of removing the antagonist and replacing it with an agonist while still retaining the rescue of the receptor to the cell surface.

To our knowledge, PC-mediated in vivo rescue of a mutant glycoprotein GPCR has not yet been demonstrated in female mice. Functional rescue of a GPCR that plays a crucial role in the reproductive axis is particularly challenging in females, owing to the innate complexity and dynamicity of the system. Furthermore, since calcium is the major signaling molecule that directly and indirectly mediates the signaling of these receptors, it is crucial to study these mutations with respect to their altered calcium responses, which has not been reported to date. $G_s$- and $G_q$-based signaling is the predominant mechanism employed by glycoprotein GPCRs and both mechanisms rely directly and indirectly on intracellular calcium levels (Brands et al, 2024; Breen et al, 2013), making calcium imaging the most robust approach to measure the signaling-related activity of these GPCRs but has not yet been employed to study the functional impact of PC treatment at the single-cell level. The pivotal role of $G_q$ signaling in triggering ovulation makes calcium imaging even more relevant for studying functional changes in the context of LHR. We further couple these findings with a demonstration of the immense potential of AI for preclinical classification of cells harboring an inactivating human GPCR mutation, paving the way for future advancements in the development of safe and effective therapeutic strategies against human diseases caused by such mutations.

Rescuing LHR function in mice bearing inactivating mutations in the *Lhcgr* gene represents a significant breakthrough. Thus far, there is no treatment option for restoring fertility in human patients harboring such mutations due to their inability to respond to exogenous LH or other treatments such as pulsatile GnRH (an upstream stimulator of LH). Although sex hormone replacement can be utilized to promote sexual development, there are no effective treatments to restore gametogenesis/fertility in these patients. We have previously demonstrated that LHR-Chap can rescue the cell surface expression of the T461I mutant human LHR, which then responds to hormone stimulation in heterologous cells

in vitro (Newton et al, 2011; Newton et al, 2021). We now demonstrate that cells from LHR-Chap-treated mice carrying the corresponding human mutation in the murine LHR acquire responsiveness to exogenous LH stimulation in ovarian slices at levels similar to those elicited in control Het ovarian slices, using functional calcium imaging. That the cells from LHR-Chap-treated Hom mice elicited an LH response similar to control Het mice indicates that no receptor desensitization (tachyphylaxis) occurred even following 30 days of chronic administration of this LHR agonist. However, using our model, we observed delayed signaling kinetics following rescue, which may be due to the lower on-rate of LHR-Chap. Differences in the physicochemical properties of the natural and synthetic ligand may provide insight into the mechanism of rescue/receptor activation by LHR-Chap. For example, these data could indicate biased and altered intracellular signaling by the LHR-Chap-rescued LHR bearing the T465I mutation. Potential alternative signaling routes downstream of the LHR will be addressed in future molecular and functional profiling studies in the LhrT465I-IRES-Cre mouse model.

Our demonstration of restoration of cyclicity in female mice harboring the equivalent of the human inactivating mutation of T461I is a proof-of-concept of the feasibility of treating infertile women with the wide range of human inactivating LHR mutations with LHR-Chap (Newton et al, 2021). Our novel preclinical mouse model provides genetic access to the affected primary cells and now sets the stage to dissect how PCs alter functional signaling and thereby recover receptor function in the affected primary cells in vivo.

It is interesting to note that no apparent effect on pubertal timing was noted in the homozygous mutant female mice, despite the effects on cyclicity. Conversely, in homozygous mutant males, fertility is unaffected. Consistent with this, these mice had apparently normal gross adult testicular morphology despite delayed puberty. In humans, a similar dimorphism is observed. Indeed, women harboring inactivating LHR variants exhibit amenorrhea/infertility, but, in contrast to males, they undergo normal pubertal development (Radomsky et al, 2024). In a previous study, male and female LHR knock-out mice both displayed altered postnatal sexual development and pubertal delay (Zhang et al, 2001), indicative of a complete absence of LHR signaling, in contrast to the current LhrT465I-IC model. This suggests that the LhrT465I-IC model retains some LHR functionality. This correlates with the impaired, but not fully abolished, cell surface expression of the murine mutant observed in vitro (Appendix Fig. S1). Indeed, the functional calcium imaging analyses indicate that there are impaired, but not abrogated and thus residual, responses to LH in the Leydig cells of the male LhrT465I-IC mice. Although no such responses were seen in female ovarian cells, this could possibly reflect differences in receptor expression and/or intracellular coupling in the two tissue types. Another possible explanation might be the effect of temperature on protein folding. Testes must maintain a 2–3 °C lower temperature than core body temperature to ensure optimal sperm production and motility (Abdelhamid et al, 2019; Bedford, 1991). This lower temperature may facilitate higher levels of spontaneous refolding of the mutant receptor into a functional state in the testes or may lead to a comparatively lower proportion of misfolded receptors in the testes compared to the ovaries ab initio (Guo et al, 2012; Zheng et al, 2024). Taken together, our data potentially reveal a sexual dimorphism in

penetrance of the murine T465I LHR mutation. This is perhaps, to some extent, unsurprising since the regulation of male and female sexual development is itself sexually dimorphic. However, the differences observed in the LHR knock out and this mutant LHR model are interesting to note and further functional investigations may be revealing with regards to the nuanced roles of LHR signaling in male/female sexual development.

Since most of the loss-of-function LHR mutations reported in humans cause intracellular retention of the receptor (Newton et al, 2016; Newton et al, 2021), PCs are uniquely positioned to potentially rescue the function of these mutant receptors. As patients with inactivating LHR mutations are hypergonadotropic due to the absence of sex steroid negative feedback, it is likely that restoring expression of these mutant receptors through increasing their cell surface expression with LHR-Chap will enable them to respond to endogenous circulating hormones. Indeed, increased functionality of several other human LHR mutants (in addition to T461I) following an increase in their cell surface expression, after LHR-Chap incubation has been demonstrated in vitro (Newton et al, 2021). This, combined with the proof-of-concept rescue of the T465I mutation in vivo, provides support for the exciting applicability of single PCs to treat multiple GPCR mutations. This is in contrast to gene editing strategies, which require a tailor-made approach for every individual mutation.

The large number of GPCRs and ever-increasing discovery of pathologies associated with inactivating mutations presents the possibility of extensive exploitation of such therapeutic strategies in the future. To aid this, we use an arsenal of existing machine-learning approaches and AI tools to formulate an automated pipeline to accurately classify cells harboring an inactivating GPCR mutation, based on subtle differences in calcium signaling profiles or signatures between mutant and control cells that cannot be detected by conventional methods. Moreover, our neural network reliably classified mutant cells that were functionally rescued using the PC, even without relying on exogenous application of the receptor agonist, identifying intrinsic features in their spontaneous calcium signaling that distinguish them from mutant, and to a much smaller extent, control cells. This classification performance was further boosted when responses to the agonist were also considered. Although the AI tools used here are themselves not methodologically novel, the novelty of this study lies in demonstrating their utility as a preclinical classifier, i.e., classification of control, mutant, and rescued primary cells from a preclinical mouse model with a human inactivating GPCR mutation, based on their intracellular calcium profiles. In future work, the performances of the AI/ML models could potentially be further enhanced by conducting a systematic parameter search or optimization procedure to identify the best-performing configurations for each algorithm. In the present study, we show that AI enables a deeper investigation of the nature of the PC-based rescue of a mutant GPCR, particularly in a heterogeneous system. An understanding of how this rescue influences downstream signaling will be important for the application of PCs to mutant GPCRs in general. We therefore posit that PC rescue of mutant GPCRs will become a viable component of personalized medicine.

LHR-Chap was originally identified as an allosteric agonist of LHR (van Koppen et al, 2008), but was later repurposed in vitro as a PC (Newton et al, 2011). LHR-Chap is an agonistic positive allosteric modulator (Ago-PAM). In addition to its positive

allosteric effect on the action of the orthosteric (endogenous) agonist, LH (Newton et al, 2011), an Ago-PAM also has agonistic properties. This agonistic activity of LHR-Chap has been shown to induce ovulation (van de Lagemaat et al, 2009), albeit with lower efficacy than LH. Therefore, LHR-Chap may cooperatively enhance the activity of endogenous circulating LH, leading to overall higher efficacy in vivo than that observed in vitro (without exogenous LH). Regardless of the contribution of agonistic functions of LHR-Chap to the observed effects, we could still demonstrate PC-based rescue of a GPCR and use AI to assess mutated cells before and after application. In comparison, the known agonist of the LHR, HCG, was not capable of rescuing the receptor. However, the differential contribution of the agonistic and chaperone properties of LHR-Chap to the functional rescue reported here needs to be investigated in a future study. We only offer a mere glimpse of the true potential of our AI model here (the current study serving as a proof-of-concept); we aim to showcase the virtually unlimited scalability and adaptive nonlinear feature-learning capabilities of our proposed model in future studies.

# Methods

### Reagents and tools table

| Reagent/resource | Reference or source | Identifier or Catalog Number |
|---|---|---|
| **Experimental models** | | |
| LhrT465I-IRES-Cre (*M. musculus*) | This study | N/A |
| eRosa26-τGFP (*M. musculus*) | Wen et al, (2011) | N/A |
| eRosa26-GCaMP3 (*M. musculus*) | Dr. D. Bergles, Johns Hopkins University, Baltimore; Paukert et al, (2014) | N/A |
| HEK 293T cell line | American Type Culture Collection (ATCC) | CRL-3216 |
| **Recombinant DNA** | | |
| pKO*Lhcgr*5′PGKneoIREScre3′ (targeting vector for mouse generation) | This study | N/A |
| AAV2/9CAGsGCaMP3_P2A_mKate2 | This study | N/A |
| pAdDeltaF6 | Addgene | #112867 |
| pAAV2/9n | Addgene | #112865 |
| CAGsGCaMP3_P2A_mKate2 | This study | N/A |
| **Antibodies** | | |
| Chicken anti-GFP | Invitrogen | A10262 (RRID: AB_2534023) |
| Goat anti-chicken Alexa 488 | Invitrogen | A11039 (RRID: AB_142924) |
| **Oligonucleotides and other sequence-based reagents** | | |
| PCR primers | This study | Methods and Protocols section |

| Reagent/resource | Reference or source | Identifier or Catalog Number |
|---|---|---|
| **Chemicals, enzymes and other reagents** | | |
| NMDG | Sigma | 66930 |
| NaCl | Grüssing GmbH | 121221000U |
| HCl | Sigma | 258148 |
| $HNO_3$ | Sigma | 225711 |
| KCl | Sigma | P9541 |
| $NaH_2PO_4$ | Roth | 4984.1 |
| $NaHCO_3$ | Roth | 6885.1 |
| HEPES | Sigma | H3375 |
| D-(+)-Glucose | Roth | X997.2 |
| Thiourea | Sigma | T7875 |
| Na-ascorbate | Sigma | 11140 |
| Na-pyruvate | Sigma | P2256 |
| $CaCl_2 \cdot 2H_2O$ | Sigma | C3306 |
| $MgSO_4 \cdot 7H_2O$ | Sigma | M2773 |
| Mayer's hematoxylin | Morphisto GmbH | 10231.00500 |
| Eosin | Morphisto GmbH | 12217.00500 |
| DePeX | Serva | 18243.02 |
| Xylol | Carl Roth | 701864 |
| Ethanol | Carl Roth | 702543 |
| HCG | Merck | C0434 |
| LHR-Chap (Org 43553/Org 42599) | CiVentiChem | N/A (custom synthesized) |
| Tissue freezing medium | Leica | 14020108926 |
| Bovine serum albumin | Sigma | A2153 |
| Donkey serum | Jackson ImmunoResearch | 017-000-121 |
| Tween-20 | Roth | 9127.1 |
| Triton X-100 | Sigma | X100 |
| DMEM | Gibco | 41966-029 |
| Fetal bovine serum | Gibco | 10270-106 |
| Matrigel matrix, growth factor reduced | BD Biosciences | 356230 |
| X-tremeGENE HP DNA transfection reagent | Roche | 6366236001 |
| Polyethylenimine | Polysciences Europe GmbH | 23966 |
| **Software** | | |
| MATLAB (version 2021b) | MathWorks | https://www.mathworks.com |
| Python (version 3.12.3) | Python Software Foundation | http://ww.python.org |
| ZEN (Black, version 14) | Zeiss | |
| Prism (version 9.5.0) | GraphPad | https://www.graphpad.com |
| ImageJ (version 1.53e) | Wayne Rasband and contributors, National Institutes of Health, USA | https://imagej.nih.gov/ij/index.html |

| Reagent/resource | Reference or source | Identifier or Catalog Number |
|---|---|---|
| **Other** | | |
| Confocal microscope | Zeiss | LSM 710 |
| Epifluorescence microscope | Zeiss | Imager.M2 |
| Slide scanner microscope | Zeiss | Axio Scan.Z1 |
| Cryostat | Leica | CM3050 S |
| Vibratome | Leica | VT1200S |
| RC-26G Imaging chamber | Warner Instruments | 640235 |
| Slice anchor for RC-26G | Warner Instruments | 640254 |
| Vacuum grease silicone | Beckman Coulter | 335148 |

## Methods and protocols

### Cell line culture

HEK 293T cells (obtained from ATCC) were maintained in complete media (DMEM supplemented with 10% FBS) at 37 °C in a humidified 5% $CO_2$ atmosphere. All cell culture plates were coated with a 1:30 dilution of Matrigel Matrix, Growth Factor Reduced (BD Biosciences) prior to cell seeding. Cells were transiently transfected with empty vector, WT or mutant LHRs using X-tremeGENE HP (1:2 ratio) DNA transfection reagent (Roche). Monthly mycoplasma testing was performed to confirm the lack of contamination.

### Quantification of mLHR expression by receptor ELISA in HEK 293 T cells

Cell surface mLHR expression was measured by receptor ELISA in intact HEK 293 T cells transiently transfected with N-terminally FLAG-tagged WT or T465I mLHR or empty vector as described previously (Newton et al, 2021).

### Generation of the LhrT465I-IRES-Cre knock-in mice

LhrT465I-IRES-Cre (LhrT465I-IC) mice were generated by homologous recombination in mouse embryonic stem (ES) cells using a targeting construct carrying the exon 11 of *Lhcgr* with the T465I mutation, followed by an internal ribosomal entry site (IRES) and a Cre recombinase sequence. This allows bicistronic expression of both a mutant LhrT465I protein and Cre recombinase (Candlish et al, 2015). The expression of Cre is thus restricted only to cells that express *Lhcgr*. Correct insertion of the targeting construct was verified using Southern blot analysis as follows. DNA was extracted from tail tip biopsies or ES cells using lysis buffer containing 0.1 mg/mL proteinase K (1 mg/mL was used for extraction from ES cells). Following extraction, genomic DNA was digested overnight with *BamHI* and run on a 0.7% agarose gel, then transferred to a nylon membrane by capillary transfer and screened by hybridization of a 508 bp $^{32}$P-labeled probe complementary to sequences located 5′ to the 5′ homology arm of the targeting construct. Probe hybridization produces a 9.5-kb band from the wild-type allele, whereas the correctly targeted allele generates a 6.5-kb band. To further confirm the integration of the T465I mutation into the targeted allele, a *BsiWI* restriction digestion site adjacent to this mutation was inserted. Cleavage by *BsiWI* confirmed the presence

of the modified DNA region containing the T465I mutation in the targeted allele. The presence of the mutation could then be confirmed by PCR in clones with the correct *BsiWI* digest pattern. Correctly targeted ES cells were injected into C57BL/6 J blastocysts to generate male chimeras that were backcrossed to C57BL/6J females to produce heterozygous LhrT465I-IC mice. Mice were then further crossed to produce a homozygous colony. Mice were maintained on a mixed genetic background of 129SvJ × C57BL/6J. The genotypes of the LhrT465I-IC mice were confirmed by PCR using the primer sequences: 5′-CTTGTGCCAACCCATTTCTG-TACG-3′ (wild-type forward primer); 5′-AAGTGTCTTGGGC-CACCCTTTG-3′ (common reverse primer) and 5′-CACGTACTGACGGTGGGAGAATG-3′ (LhrT465I-IC allele reverse primer). WT offspring were confirmed by the presence of a single band of 394 bp. For the LhrT465I-IC allele, heterozygous offspring gave two products of 394 and 697 bp, whereas homozygous offspring were identified by the presence of one band at 697 bp.

### Mice

Mice were housed under a standard light/dark cycle with food and water ad libitum. To label Lhr-expressing cells, LhrT465I-IC mice were crossed with eROSA26-τGFP (GFP) (Wen et al, 2011) or eROSA26-GCaMP3 (GCaMP) (Paukert et al, 2014) (generously provided by Dr. D. Bergles, Johns Hopkins University, Baltimore) animals. In the resulting Lhr-reporter/effector mice, Cre recombinase is expressed under the control of the *Lhcgr* promoter. Cre-mediated recombination results in the removal of a strong transcriptional stop cassette (three SV40 polyA signals) from the ROSA26 locus and subsequent constitutive reporter expression in Lhr-expressing cells. Mice were kept in a mixed (129/SvJ and C57BL/6 J) background. All Lhr-reporter/effector mice used in this study were heterozygous for the eR26-reporter/effector alleles.

### Tissue preparation for immunohistochemistry

Animals were anaesthetized with an i.p. injection of ketamine/xylazine, and transcardially perfused with 1× PBS followed by 4% paraformaldehyde in PBS. Reproductive organs were removed and post-fixed for 2 h in 4% paraformaldehyde in PBS at 4 °C. Following overnight incubation in 18% sucrose in PBS, tissues were frozen in tissue freezing medium (Leica, Nussloch, Germany) in a dry ice/ethanol slurry. 14-μm-thick sections were cut using a cryostat (Leica), and stored at −80 °C until use.

### Immunohistochemistry

Sections were washed in PBS, and incubated in 10% donkey serum, 3% bovine serum albumin, 0.3% Triton X-100 in 1× PBS for 1 h. Sections were then incubated with antibodies against GFP (chicken; 1:2000; A10262; Invitrogen, Carlsbad, CA, USA; RRID: AB_2534023) overnight at 4 °C. Following washing with 0.05% Tween-20 in 1× PBS, sections were incubated with goat anti-chicken Alexa 488 (1:400; A11039; Invitrogen; RRID: AB_142924) for 2 h at room temperature. Cell nuclei were counterstained with bisbenzimide. Images were captured using a Zeiss Axio Scan.Z1 slide scanner (Zeiss, Jena, Germany).

### Hematoxylin and eosin (H&E) staining

Cryosections were washed for a short duration in 1× PBS to rehydrate the tissue, followed by a washing step in ddH$_2$O. The

sections were then immersed in xylol (2×, 5 min each) and rehydrated by sequential incubation in an alcohol gradient (100, 96, 80, and 75%; 5 min each). The sections were kept in tap water for 5 min, dried, and then stained with Mayer's hematoxylin (Morphisto) for 5 min, rinsed in lukewarm running tap water for 5 min, incubated for 5 min in eosin (Morphisto), and dehydrated by sequential incubation in an alcohol gradient (3× 75%, 3× 80%, 3× 96%, and 3× 100%). The sections were then washed twice in xylol and mounted with DePeX (Serva).

### HCG and LHR-Chap administration

HCG and LHR-Chap were systemically administered by i.p. injection. HCG (Merck, C0434) was injected at a concentration of 500 IU/kg. This dose has been shown to induce ovulation with 100% efficacy in primed WT mice (van de Lagemaat et al, 2009). Oral administration of 50 mg/kg LHR-Chap (custom synthesized by CiVentiChem), also referred to as Org 43553 or Org 42599 in previous studies, has also been shown to induce ovulation in mice (van de Lagemaat et al, 2009; van Straten et al, 2002). We reasoned that intraperitoneal administration would be more efficient even at a lower dose. Moreover, long-term administration would be required to maintain steady-state levels of the chaperone in the Lhr-expressing cells so that the cells are ready to respond to the natural LH surge, when it occurs. Our previous in vitro study demonstrates that LHR-Chap needs at least 24 h to cause an optimal increase in cell surface Lhr expression (Newton et al, 2011). This duration is likely to be in the 24–48 h range in vivo. Therefore, we administered 25 mg/kg LHR-Chap (i.p.) for 3 consecutive days, followed by every alternate day until the experimental endpoint. The same treatment paradigm was followed for HCG treatment, as a control.

### Estrous cycle staging

To monitor estrous cycles, vaginal fluid was collected from adult homozygous mutant females and their WT and heterozygous littermates. Smearing of the vaginal fluid and subsequent observation of vaginal cytology under a brightfield microscope revealed the estrous cycle stage (Caligioni, 2009). Vaginal smear data were collected for at least 2 weeks to assess the regularity of the estrous cycle for each female.

### Breeding tests and fertility index

To evaluate natural fertility, at least 8–10-week-old female mice were individually and continuously mated with one male mouse (at least 8-week-old) (Handelsman et al, 2020; Walters et al, 2007). The fertility index was calculated as described previously (Handelsman et al, 2020). Briefly, the fertility index for each individual breeding pair is defined as the regression slope of the cumulative number of pups born over the elapsed time of the mating trial. This provides a single estimate of fertility per mouse over the whole mating trial, taking into account litter frequency and size.

### AAV vector production

The AAV2/9CAGsGCaMP3_mKate2 (AAV9-GCaMP3) virus was produced using the triple transfection helper-free method. This involved transfecting HEK293T cells in culture with three plasmids in a 1:1:1 ratio, the first containing essential viral genes such as E2 and E4 (pAdDeltaF6 was a gift from James M. Wilson; Addgene plasmid #112867), the second which facilitates generation of serotype 9 AAV vectors (pAAV2/9n was a gift from James M. Wilson; Addgene plasmid #112865), and the third, which dictates the packaged contents of the virus particles, was cloned such that the GCaMP3 sequence was driven by the CAGs promoter and followed by WPRE and SV40 polyA sequences. Transfection was undertaken when the cells reached 60–70% confluence using a 4:1 (v:w) ratio of polyethylenimine (PEI) (Polysciences Europe GmbH) to plasmid DNA. About 60–72 h after transfection, cells were pelleted and processed to recover the virus.

### Micro-injection of AAV-GCaMP3

The right and left flanks of the mice were carefully shaved, after which a small incision was made, cutting through the skin, subcutaneous fat, and muscle layer. The fat pad and surrounding tissue is gently lifted outwards to exteriorize the gonads through the incision. AAV9-GCaMP3 was slowly microinjected into the ovaries of anesthetized female WT (LhrT465I-IC$^{-/-}$) and Het (LhrT465I-IC$^{+/-}$) mice (2 μl per ovary). The gonads are then carefully placed back into the abdominal cavity. The muscle layer is sutured, followed by the skin. The microinjected mice were anesthetized and transcardially perfused with room temperature (RT) NMDG-HEPES buffer after 3–4 weeks at the estrus stage of the estrous cycle, and the reproductive organs were harvested for calcium imaging.

### Acute ovary and testis slice preparation and calcium imaging

The protocol for preparing acute tissue slices was modified from (Ting et al, 2018). Briefly, adult LhrT465I-IC$^{+/-}$/eR26-GCaMP (Het) and LhrT465I-IC$^{+/+}$/eR26-GCaMP (Hom) female mice at the estrus stage of the estrous cycle were deeply anesthetized with a mix of ketamine/xylazine and transcardially perfused with RT NMDG-HEPES buffer (containing in mM: NMDG 92, HCl 92, KCl 2.5, NaH$_2$PO$_4$ 1.2, NaHCO$_3$ 30, HEPES 20, glucose 25, thiourea 2, Na-ascorbate 5, Na-pyruvate 3, CaCl$_2$•2H$_2$O 0.5, and MgSO$_4$•7H$_2$O 10). For slicing, the ovary or testis was dissected out and transferred to RT NMDG-HEPES buffer continuously aerated with 95% O$_2$/5% CO$_2$. 200- and 400-μm thick slices were obtained from ovaries and testes, respectively, using a Leica VT1200S vibratome. The slices were transferred and incubated for 10–12 min in pre-warmed (34 °C) NMDG-HEPES buffer for recovery. The slices were then transferred to HEPES holding buffer (containing in mM: NaCl 92, KCl 2.5, NaH$_2$PO$_4$ 1.2, NaHCO$_3$ 30, HEPES 20, glucose 25, thiourea 2, Na-ascorbate 5, Na-pyruvate 3, CaCl$_2$•2H$_2$O 2, and MgSO$_4$•7H$_2$O 2) at RT with constant carbogenation. Following this, the sections were kept in aerated HEPES holding buffer for at least 30 min before imaging. Tissue slices were constantly perfused by recording buffer (containing in mM: NaCl 124, KCl 2.5, NaH$_2$PO$_4$ 1.2, NaHCO$_3$ 24, HEPES 5, glucose 12.5, CaCl$_2$•2H$_2$O 2, and MgSO$_4$•7H$_2$O 2) during imaging. Imaging was performed with a Zeiss LSM 710 confocal imaging system operated by the ZEN software (Carl Zeiss ZEN 2012 [black] 64 bits, Version 14). To hold the tissue slice, an RC-26G Open Diamond Bath Imaging Chamber (Warner Instruments) was routinely used. For the chamber floor, glass coverslips (22 × 40 mm, CS-22/40, Warner Instruments) were prepared by soaking them overnight in nitric acid (70%, Sigma Aldrich) and then washing with distilled water until reaching pH ~7.0. The coverslips were attached to the chamber with vacuum grease silicone (Beckman Coulter, Cat. No. 335148). The tissue slices were then transferred into the chamber and attached to the

floor by short solution removal followed by replenishment of the buffer in the chamber. The solution flow rate through the chamber was kept at 2 ml/min using a custom-made tube perfusion system. During imaging, all bath-applied solutions were aerated (95% $O_2$/ 5% $CO_2$). Imaging was performed using a 20× water immersion objective (W Plan-Apochromat 20×/1.0 DIC VIS-IR, Zeiss). The frequency of frame collection was set to 2 Hz. The total imaging time was 12.5 min; since two images were captured per second, a total of 1500 images were obtained. A maximum intensity projection of the recorded images was used for manually marking ROIs (using the freehand selection tool on ImageJ). ROIs were marked by two independent human experts blinded to the experimental groups, with minimal inter-rater variability. A total of 56 time-series confocal recordings (each comprised of 1500 images) were used, with a total of 1946 ROIs (cells) being annotated. Calcium imaging corrections and statistical analyses were done with MATLAB_2021b (Mathworks, Natick, MA, USA). Heatmaps for the fluorescence responses were made using MATLAB.

### AI-assisted analyses

The experimenters performing the mathematical modeling and analyses were fully blinded to the experimental details as well as the identities of the experimental groups. All analyses were performed using Python (Python Software Foundation, Python Language Reference, version 3.12.3. Available at http://ww.python. org). For the clustering analyses, each cell's unique time-series calcium trace/signature ($\Delta F/F_0$ over 1500 time-points) was treated as a high-dimensional vector in $\mathbb{R}^{1500}$, with each time-point representing one dimension (1500 dimensions in total). These vectors were reduced using t-SNE (and other classical machine-learning methods) to a 2D embedding that groups cells according to similarity in their calcium-signaling dynamics alone; spatial location and morphology were not included in this analysis. The t-SNE algorithm projects these data onto a 2D space while preserving local relationships between cells, such that nearby points in the original space remain close in the embedding (Van der Maaten and Hinton, 2008). Formally, t-SNE minimizes the Kullback-Leibler divergence (Kullback and Leibler, 1951) between two probability distributions: one measuring pairwise similarity in the original high-dimensional space, and the other in the low-dimensional embedding.

Let $\mathbf{x}_i \in \mathbb{R}^{1500}$ denote the signal from cell $i$. The algorithm constructs conditional probabilities $p_{j|i}$ representing similarities between cells based on a Gaussian kernel centered at $\mathbf{x}_i$. These are symmetrized to yield $p_{ij}$, the joint probability of similarity. A corresponding distribution $q_{ij}$ is computed in the 2D map using a Student's $t$-distribution. The embedding is optimized by minimizing:

$$\mathrm{KL}(P\|Q) = \sum_{i \neq j} p_{ij} \log\left(\frac{p_{ij}}{q_{ij}}\right)$$

This process ensures that similar calcium response patterns are grouped together in the resulting 2D map, facilitating intuitive visualization of cellular differences. For this analysis, we used the scikit-learn implementation of t-SNE with two output dimensions, with a fixed random seed 42 to ensure reproducibility. The perplexity used is 30. This parameter reflects the effective number of nearest neighbors considered when computing the

high-dimensional similarity distribution, balancing attention between local and global data structure. We run the optimization using gradient descent to minimize the KL divergence for 1000 steps.

**Algorithm 1**: **K-means clustering on t-SNE-embedded data**
Let $X = \{\mathbf{x}_1, \mathbf{x}_2, \ldots, \mathbf{x}_n\} \in \mathbb{R}^2$ represent the t-SNE-embedded calcium trace dataset, where each cell is represented by a 2D vector. K denotes the number of desired clusters which varies from 2 to 5. $\boldsymbol{\mu}_k \in \mathbb{R}^2$ denotes the centroid of cluster $k$, for $k = 1, \ldots, K$ and $C_k \subseteq X$ be the set of data points assigned to cluster $k$. The algorithm proceeds as follows:

1. **Initialization:** Randomly select K initial centroids $\boldsymbol{\mu}_1, \ldots, \boldsymbol{\mu}_K \in \mathbb{R}^2$
2. **Assignment step:** For each data point x_i, assign it to the nearest centroid:

$$c_i = \arg \min_k \|\mathbf{x}_i - \boldsymbol{\mu}_k\|^2.$$

3. **Update step:** Recalculate each centroid as the mean of all points assigned to that cluster:

$$\boldsymbol{\mu}_k = \frac{1}{|C_k|} \sum_{\mathbf{x}_i \in C_k} \mathbf{x}_i.$$

4. **Convergence:** Repeat the assignment and update steps until cluster assignments no longer change or a predetermined maximum number of iterations is reached.

### K-means clustering

K-means clustering is an unsupervised learning algorithm that requires the number of clusters K to be specified in advance (MacQueen, 1967). We evaluate for K values 2, 3, 4, and 5. For each cluster we evaluate how homogeneous the cluster is with respect to the biological type of the points in the cluster. That is, we compare the homogeneity of the Het (control) and Hom (mutant) groups in the clusters discovered by the algorithm. To evaluate clustering quality, we used the silhouette score, which measures how similar each point is to its own cluster compared to other clusters. Scores closer to 1 indicate well-separated and cohesive clusters. The silhouette score for a point $i$ is given by:

$$s(i) = \frac{b(i) - a(i)}{\max(a(i), b(i))},$$

where $a(i)$ is the average distance between point $i$ and all other points in its own cluster, and $b(i)$ is the minimum average distance between $i$ and all points in the nearest neighboring cluster. Scores closer to 1 indicate well-separated and cohesive clusters. For a cluster, the silhouette score is the mean of the silhouette scores of all points in that cluster.

### Neural network architecture and training procedure

Each calcium trace is represented as a 1500-dimensional input vector corresponding to the full time-series of $\Delta F/F_0$ measurements. The network architecture consisted of an input layer, two fully connected hidden layers (512 and 128 units, respectively) with

ReLU activation and staged dropout ($p = 0.5$ after the first hidden layer and $p = 0.2$ after the second hidden layer), and an output layer with softmax activation for classification.

Formally, the model $\theta$ maps an input trace $x \in \mathbb{R}^{1500}$ through

$$z_1 = \mathrm{ReLU}(W_1 x + b_1),$$

$$z_1 \rightarrow \mathrm{Dropout\ with\ } p = 0.5,$$

$$z_2 = W_2 z_1 + b_2 \quad (128\text{-dimensional pre-activation feature}),$$

$$z_2 \rightarrow \mathrm{ReLU, then\ Dropout\ with\ } p = 0.2,$$

$$y = W_3 h + b_3,$$

followed by:

$$p = \mathrm{Softmax}(y)$$

for classification.

We used the PyTorch framework for implementation (module structure:

Linear($1500 \rightarrow 512$) $\rightarrow$ ReLU $\rightarrow$ Dropout($0.5$) $\rightarrow$ Linear($512 \rightarrow 128$) $\rightarrow$
ReLU $\rightarrow$ Dropout($0.2$) $\rightarrow$ Linear($128 \rightarrow \mathrm{output\_size}$) $\rightarrow$ Softmax($\mathrm{dim} = 1$)).

The network was trained using a cross-entropy loss function and optimized with the Adam optimizer (learning rate = 0.001). Early stopping was employed with a patience of 30 epochs to prevent overfitting, and training was terminated if the validation accuracy did not improve during this window. A batch size of 32 was used, and the dataset was split into training and testing sets in an 80:20 ratio, stratified by class. Each model was trained independently on male and female datasets. For genotype classification, we used output_size = 2 (Het vs. Hom). After training, the best-performing model (based on validation accuracy) was saved and used to extract intermediate hidden-layer features from all samples. These 128-dimensional features (the pre-activation output of the second hidden layer, $z_2$) serve as a compressed representation of the calcium trace and were used for downstream visualization and interpretation.

For analyzing the "basal" (before LH application) part of the calcium imaging time-series data, we used only the first 600 frames (corresponding to the first 5 min).

### Feature space representation

The trained network defines a transformation from the original time-series data into a lower-dimensional feature space where the separation between biological classes is optimized for classification. Since the model is explicitly trained to distinguish between control Het and Hom calcium signals, this learned feature space captures discriminative patterns that best explain class-level differences in calcium dynamics.

To visualize the 128-dimensional space and assess how well different cell types separate, we passed all samples, including LHR-Chap-treated cells, through the trained model and extracted their hidden-layer representations ($z_2$). These features were then projected into 2D using t-SNE. The key intuition is that if LHR-Chap-treated Hom cells adopt control Het-like signaling patterns, their feature embeddings should cluster more closely with control Het cells in this trained, discriminative space, despite not being included in training. In contrast, cells retaining untreated mutant-like features will remain near the Hom cluster. By examining the location of each group within this feature space, we can test the hypothesis that LHR-Chap treatment shifts the signaling phenotype of Hom cells closer to that of control Het cells.

### AI image analysis

To extend our analysis beyond pre-defined cell traces, we implemented a complementary, image-based approach to identify regions of interest (ROIs) directly from raw confocal microscope images. The goal was to determine whether a trained AI model could autonomously identify ROIs that are predictive of genotype (control Het vs. mutant Hom), and whether these regions correspond to biologically relevant cells, as selected by a human expert. This allows us to assess both the classification performance of the model and the biological plausibility of its attention. This is an initial exploration, as we do not fully leverage the possibility of an image-based analysis beyond identifying ROIs for creating traces. This was deemed to be the most suitable approach, avoiding possible confounding factors and enhancing interpretability, within the current scope, since we intended this preclinical classifier to rely only on changes in calcium signals. A more complete analysis, maximizing the potential of utilizing all the information contained in the images, will be performed in a future study.

We used a B-Cos-transformer architecture (Böhle et al, 2022; Dosovitskiy et al, 2021), an explainable vision transformer framework that produces built-in saliency maps without requiring post hoc interpretation methods. The model was trained on full-field images from control Het and mutant Hom samples, using only the image-level label (Het or Hom) and without explicit annotation of cell locations. Once trained, the model was able to achieve 73% accuracy in distinguishing between the two classes on the test dataset comprising 20% of the samples stratified for the two experimental groups. This indicates that informative visual features exist at the level of image context and cellular organization.

### Classifier benchmarking

To benchmark the performance of our neural network classifier, we compared it with a panel of widely used machine-learning algorithms, each of which detects patterns in different ways. Support vector machines (SVMs) learn to separate groups of samples by drawing an optimal boundary (or "margin") between them; a linear kernel identifies straight-line separations, whereas the radial basis function (RBF) kernel captures more complex, curved boundaries. Logistic regression is a simpler method that models the probability of each sample belonging to a class, making it effective when differences between groups can be explained by additive contributions of features.

Decision Trees take a different approach: they split the data step by step based on the most informative features, producing an interpretable set of rules. Random Forests extend this idea by combining many such trees, which reduces noise and better captures variability, making them robust to heterogeneous data. K-Nearest Neighbors (KNN) instead relies on similarity: a sample

is classified based on the majority label of its closest neighbors, making it particularly sensitive to local patterns in the data. Gaussian Naïve Bayes assumes features are normally distributed and independent, a simple bias that can work surprisingly well when the data roughly match these assumptions.

Finally, the multi-layer perceptron, a type of neural network, can learn flexible combinations of features by passing data through successive "hidden layers." Each layer detects increasingly abstract patterns, allowing the model to capture nonlinear and subtle differences in calcium signaling. Together, these methods provide complementary perspectives: some favor simple global patterns (logistic regression, linear SVM), while others emphasize local structure (KNN, RBF SVM), hierarchical rules (decision trees, random forests), or more adaptive, layered representations (multi-layer perceptron).

Baselines (scikit-learn, default unless specified):

SVM (RBF): kernel = "rbf", C = 1.0, gamma = "scale"

SVM (Linear): kernel = "linear", C = 1.0

Logistic regression: penalty = "l2", C = 1.0, solver = "lbfgs", max_iter = 1000

Decision Tree: criterion = "gini", max_depth = None, min_samples_split = 2, min_samples_leaf = 1

K nearest neighbor: n_neighbors = 5, weights = "uniform", metric = "minkowski", $p = 2$

Gaussian Naïve Bayes: var_smoothing = 1e-9

Random Forest: n_estimators = 50, criterion = "gini", max_depth = None, max_features = "sqrt"

Multi-layer perceptron: hidden_layer_sizes = (100,), activation = "relu", solver = "adam", alpha = 1e-4, max_iter = 200

### Statistics

Heatmaps from the calcium imaging data were prepared using MATLAB 2021b (Mathworks, Natick, MA, USA). The data were collected and analyzed offline. Statistical tests were performed using GraphPad Prism version 9.5 (San Diego, CA, USA). No samples or animals were excluded from analyses. Comparisons between groups were done with Student's $t$-test or one- or two-way ANOVA, followed by Bonferroni or Tukey's post hoc test. For dot plots, data represent mean ± SEM. For box-and-whisker plots, first and third quartiles are shown by the box edges, and the median is shown by the solid line inside the box; the whiskers indicate the spread of the data. A $P$ value less than 0.05 was considered significant.

All AI-assisted and traditional machine-learning statistical analyses were performed using Python (Python Software Foundation, Python Language Reference, version 3.12.3. Available at http://ww.python.org).

Sample size was determined empirically based on our previous experience with experimental variability. All experiments were performed with at least three biological replicates. No randomization was performed to allocate animals into experimental groups; littermates were allocated to different experimental groups purely based on their genotype. Within the same genotype (e.g., Hom mutant mice), animals were randomly assigned to the experimental treatments (e.g., HCG vs LHR-Chap administration).

### Study approval

Animal care and experimental procedures were approved by the Animal Welfare Committee of Saarland University (31/2020 and 12/2025) and were performed in accordance with their established

---

**The paper explained**

**Problem**

GPCR mutations underlie many human diseases. Existing small molecule ligands, repurposed as pharmacological chaperones (PCs), could restore mutant GPCR folding and function, offering tantalizing therapeutic potential. However, in vivo PC-driven functional rescue of a mutant glycoprotein GPCR has not yet been shown in female mice.

**Results**

Our novel mouse model harboring a human mutation in the luteinizing hormone receptor exhibited irregular estrous cyclicity, complete absence of ovulation, and strongly diminished calcium responses to the hormone, which were all rescued/restored upon chronic systemic administration of a known synthetic ligand, repurposed here as a PC. Artificial intelligence uncovered high intrinsic heterogeneity in calcium profiles of target cells, detecting differences in the signals between the control, mutant, and rescued cells, even before LH was applied.

**Impact**

Our study provides a proof-of-concept for the feasibility of using PCs for rescuing mutant GPCR function and also demonstrates the utility of artificial intelligence in coherent interpretation of complex systems to determine the preclinical therapeutic potential of PCs.

---

guidelines. Experiments were designed based on accepted standards of animal care, and all efforts were made to minimize animal suffering.

## Data availability

The computer code, AI/ML results, and source data produced in this study are available in the following database (GitHub): https://github.com/lhranalysis/LHR_Analysis. Source data for Fig. EV2 is available on BioImage Archive, under accession number S-BIAD2345.

The source data of this paper are collected in the following database record: biostudies:S-SCDT-10_1038-S44321-025-00369-2.

## Peer review information

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

## Acknowledgements

The authors thank Dr. Igor Gamayun and Dr. Samer Alasmi for technical assistance. This project was supported by the Deutsche Forschungsgemeinschaft (DFG) through grants to UB (SFB/TR 152 and SFB 894), the Harry Oppenheimer Trust through an award to RM, the South African National Research Foundation (NRF) through grants to RM (105824) and CN (94008), and COST action BM1105 to UB and RM.

## Author contributions

**Debajyoti Das**: Conceptualization; Formal analysis; Investigation; Visualization; Methodology; Writing—original draft; Writing—review and editing. **Amanda Wyatt**: Writing—original draft; Writing—review and editing; Generation of animal model and AAV. **Sarath Sivaprasad**: Data curation; Software; Formal analysis; Visualization; Writing—review and editing. **Vanessa Wahl**: Validation; Investigation; Generation of AAV. **Sen Qiao**: Formal analysis; Investigation. **Fabien Ectors**: Generation of animal model. **Zulfiah M Moosa**: Formal analysis; Investigation; In vitro expression of the mutant receptor. **Claire L Newton**: Resources; Supervision; Funding acquisition; Writing—review and editing. **Mario Fritz**: Supervision; Writing—review and editing. **Robert P Millar**: Conceptualization of pharmacological rescue; Resources; Funding acquisition; Writing—review and editing. **Ulrich Boehm**: Conceptualization; Resources; Supervision; Funding acquisition; Writing—original draft; Writing—review and editing.

Source data underlying figure panels in this paper may have individual authorship assigned. Where available, figure panel/source data authorship is listed in the following database record: biostudies:S-SCDT-10_1038-S44321-025-00369-2.

## Funding

## Disclosure and competing interests statement

The authors declare no competing interests.

# Expanded View Figures

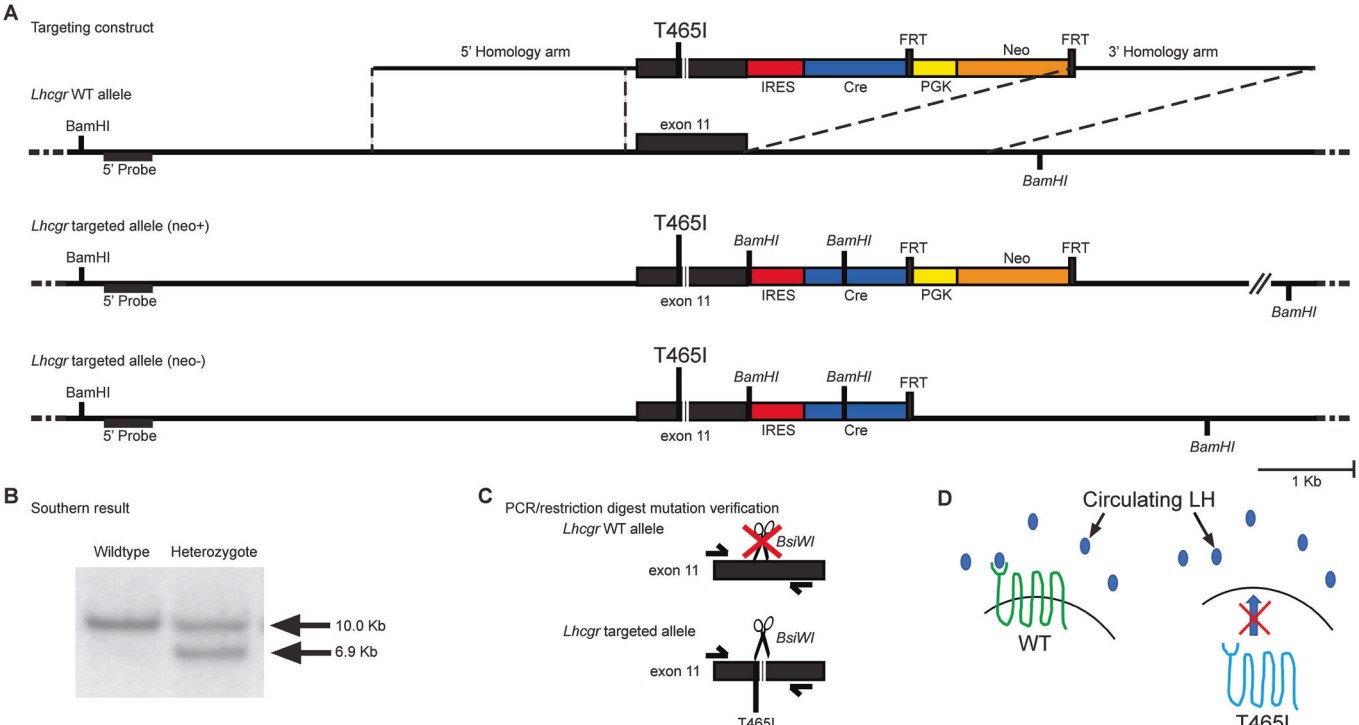

**Figure EV1. A novel mouse strain harboring the LHRT465I mutation and driving Cre recombinase expression under the control of the *Lhcgr* promoter.**

(A) A schematic diagram of the targeting construct used for the generation of the LHRT465I-IC$^{+/-}$ mouse model. The targeting construct carries the T465I mutation in exon 11, followed by an IRES and a Cre recombinase-encoding sequence (Candlish et al, 2015). (B) Southern blot showing the two bands obtained for the heterozygote mouse upon cleavage of the targeted allele by Bam*HI*. (C) A Bsi*WI* restriction digestion site adjacent to the T465I mutation was introduced into exon 11 to confirm its integration in the targeted allele. (D) A schematic diagram of the effect of the T465I mutation. The wild-type (WT) LHR is a G protein-coupled receptor that is targeted to the cell membrane, where it can bind circulating LH. However, the T465I mutation impairs recruitment of the mutant LHR to the cell membrane, preventing it from binding and responding to circulating LH. LH luteinizing hormone, IRES internal ribosomal entry site. Source data are available online for this figure.

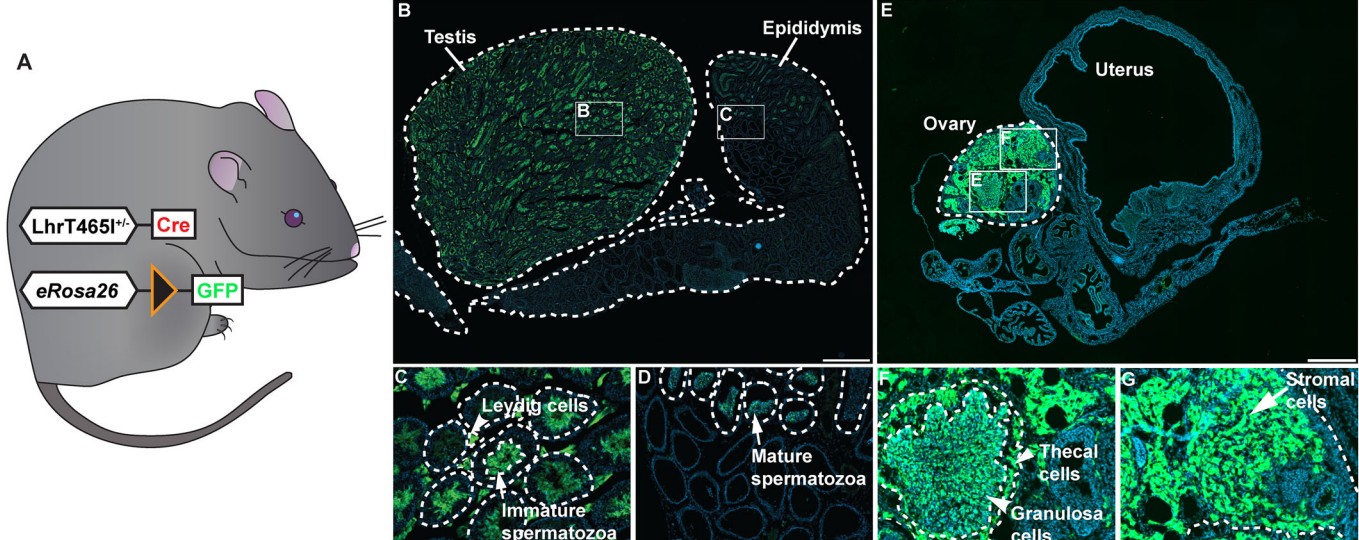

**Figure EV2.  Reporter gene expression in the reproductive organs of the LhrT465I-IC$^{+/-}$/eR26-GFP mice.**

(**A**) LhrT465I-IC$^{+/-}$/eR26-GFP mice were generated by crossing LhrT465I-IC$^{+/-}$ mice with eR26-GFP mice. In these mice, all cells expressing *Lhcgr* also express Cre, which excises the stop signal upstream of GFP. Hence, all cells expressing the luteinizing hormone receptor (LHR) also express GFP. (**B**) Representative image of a testis from a LhrT465I-IC$^{+/-}$/eR26-GFP mouse showing GFP expression in the Leydig cells and spermatozoa. Scale bar: 2000 µm. (**C**) Inset showing GFP expression in Leydig cells (triangle arrow) and mature spermatozoa (arrow) inside seminiferous tubules. Scale bar: 200 µm. (**D**) Inset showing GFP expression in immature spermatozoa (arrow) inside the epididymis. Scale bar: 200 µm. (**E**) Representative image of an ovary from a LhrT465I-IC$^{+/-}$/eR26-GFP mouse showing GFP expression in the granulosa, thecal and stromal cells. Scale bar: 500 µm. (**F**) Inset showing GFP expression in granulosa (triangle arrow) and thecal (small triangle arrow) cells. Scale bar: 50 µm. (**G**) Inset showing GFP expression in stromal cells. Scale bars: 50 µm.

## Unsupervised clustering (male mice)

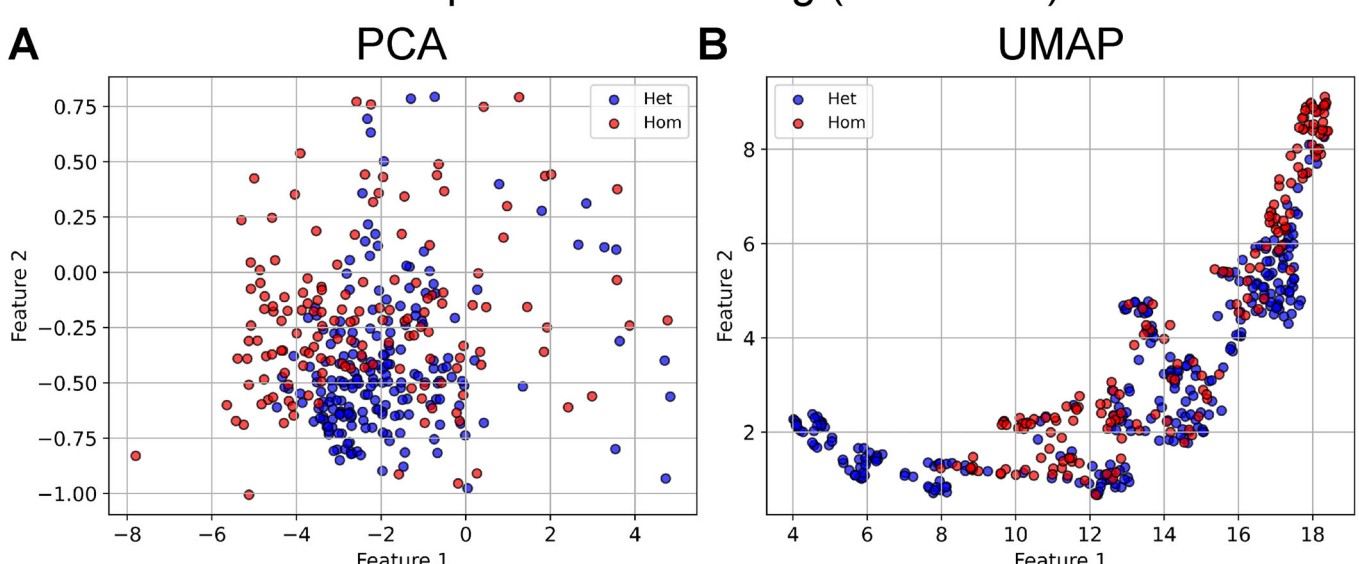

## Supervised clustering (male mice)

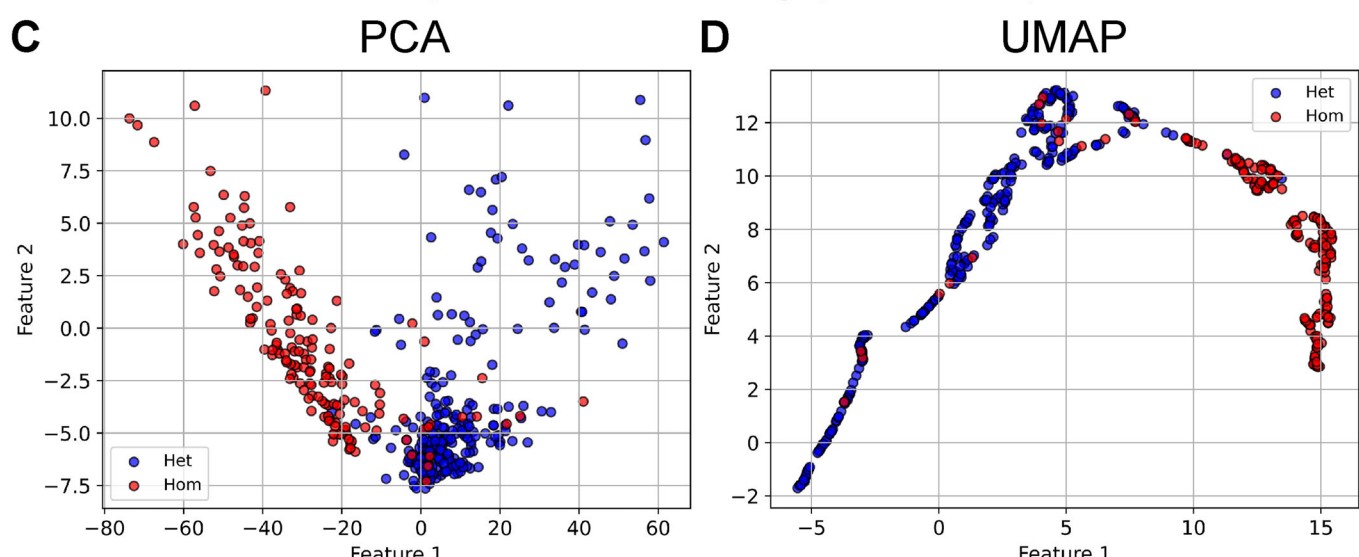

**Figure EV3.  PCA and UMAP plots for visualizing the distribution of Leydig cells based on their calcium profiles.**

(A) A PCA plot for unsupervised clustering (without AI), which shows patterns in the direction of highest variance, does not clearly distinguish between Leydig cells from control Het and mutant Hom male mice. (B) A UMAP plot for unsupervised clustering, which best preserves the true global geometry, is slightly better at distinguishing between Het and Hom cells, with some small subclusters beginning to emerge. (C) PCA representation of supervised clustering (AI model) shows two distinct clusters, predominantly composed of control Het or mutant Hom cells. (D) UMAP representation of supervised clustering shows a narrower distribution of cells, with tighter, largely pure subclusters.

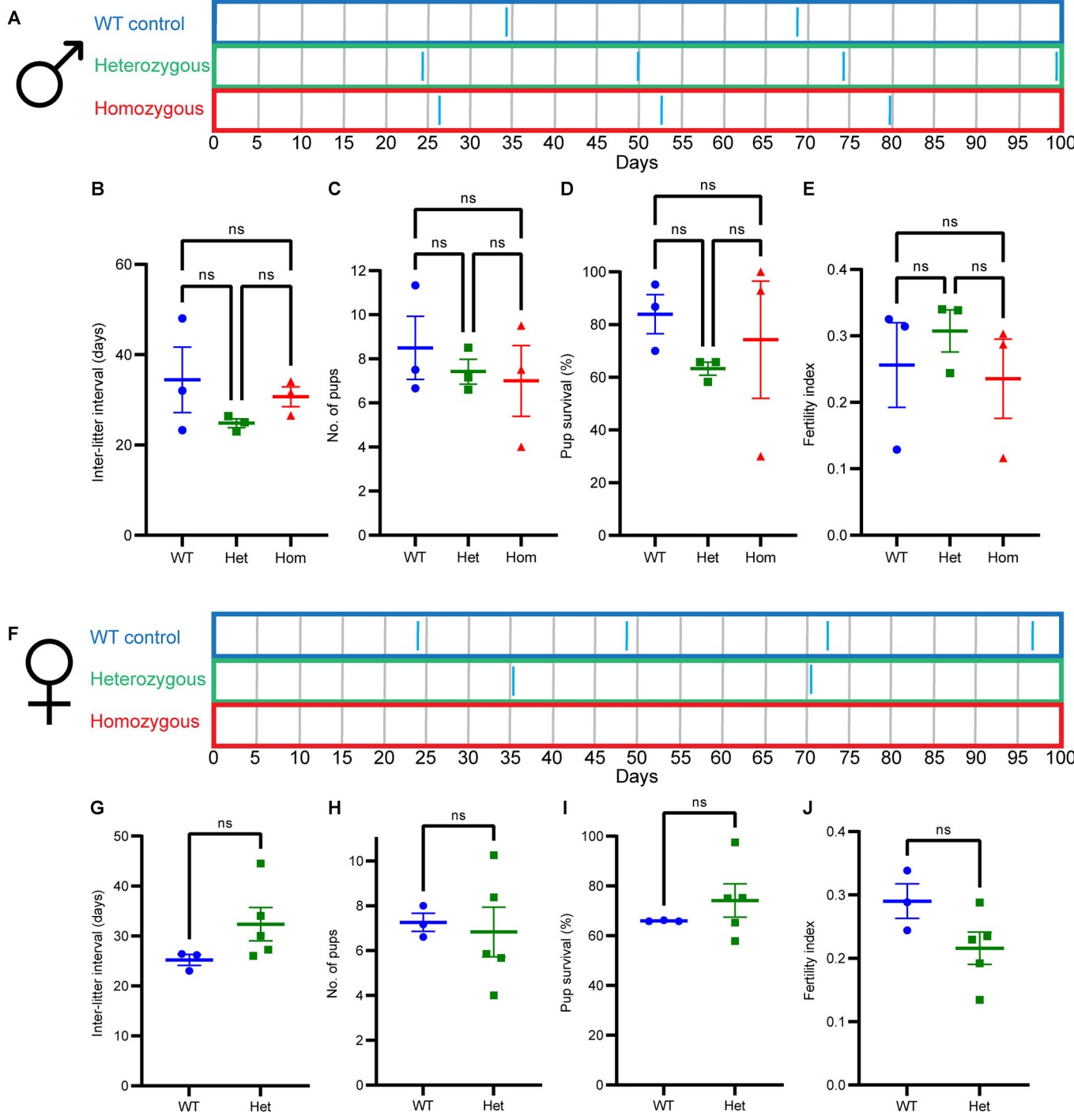

**Figure EV4. Homozygous mutant females (LhrT465I-IC$^{+/+}$), but not males, are infertile.**

(A) Breeding pairs were set up for WT control (LhrT465I-IC$^{-/-}$), heterozygous (Het) (LhrT465I-IC$^{+/-}$), and homozygous (Hom) male mice with one WT female mouse each; representative breeding pairs (with WT mate) have been shown, with blue ticks marking the birth of a litter. Based on the number of days elapsed before the birth of each litter, all three groups showed comparable fertility. (B) Time elapsed between two litters (or until birth of the first one) was plotted as inter-litter interval. ns indicates lack of statistical significance ($P = 0.3384$ for WT vs. Het, $P = 0.8246$ for WT vs. Hom, $P = 0.6369$ for Het vs. Hom). (C) Average number of pups born per litter was similar across groups ($P = 0.8292$ for WT vs. Het, $P = 0.7029$ for WT vs. Hom, $P = 0.9709$ for Het vs. Hom). (D) The survival rate of pups was also not significantly different across the groups ($P = 0.5622$ for WT vs. Het, $P = 0.8719$ for WT vs. Hom, $P = 0.8404$ for Het vs. Hom). (E) Fertility index is a composite measure considering the rate of birth of pups and litter size; this index was also similar across groups ($P = 0.7842$ for WT vs. Het, $P = 0.9604$ for WT vs. Hom, $P = 0.6318$ for Het vs. Hom). (F) Breeding pairs were set up for WT control, heterozygous, and homozygous female mice with one WT (or Het) male mouse each; representative breeding pairs have been shown. WT and Het females exhibited similar breeding success, whereas the Hom mutant female mice were infertile. (G) Inter-litter interval was not significantly different between WT and Het females ($P = 0.1638$). (H) The average number of pups per litter was not significantly different between WT and Het females ($P = 0.7863$). (I) Survival rate of pups was not significantly different between WT and Het females ($P = 0.3909$). (J) The composite fertility index, although showing a downward trend for Het females compared to WT females, was not significantly different between the two groups ($P = 0.1076$). Data information: In (B–E), data were presented as mean ± SEM. *$P < 0.05$ (ANOVA). $N = 3$ mice per group. In (G–J), data were presented as mean ± SEM. *$P < 0.05$ (two-tailed unpaired Student's $t$-test). $N = 3$ (WT) and 5 (Het). Source data are available online for this figure.

 

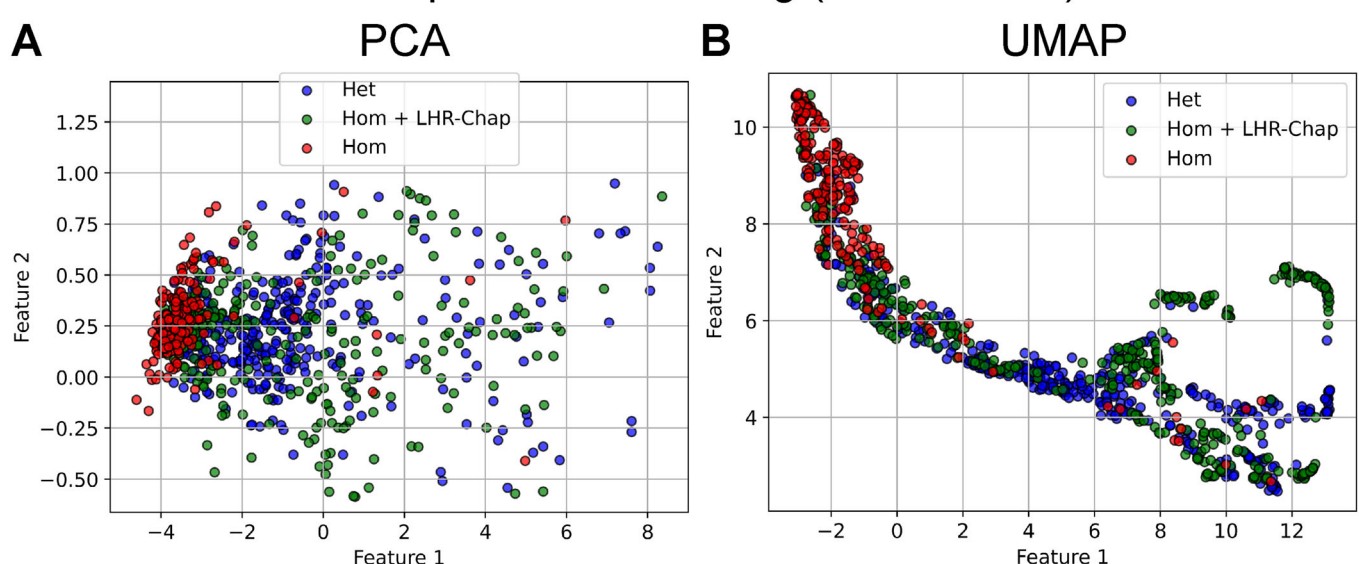

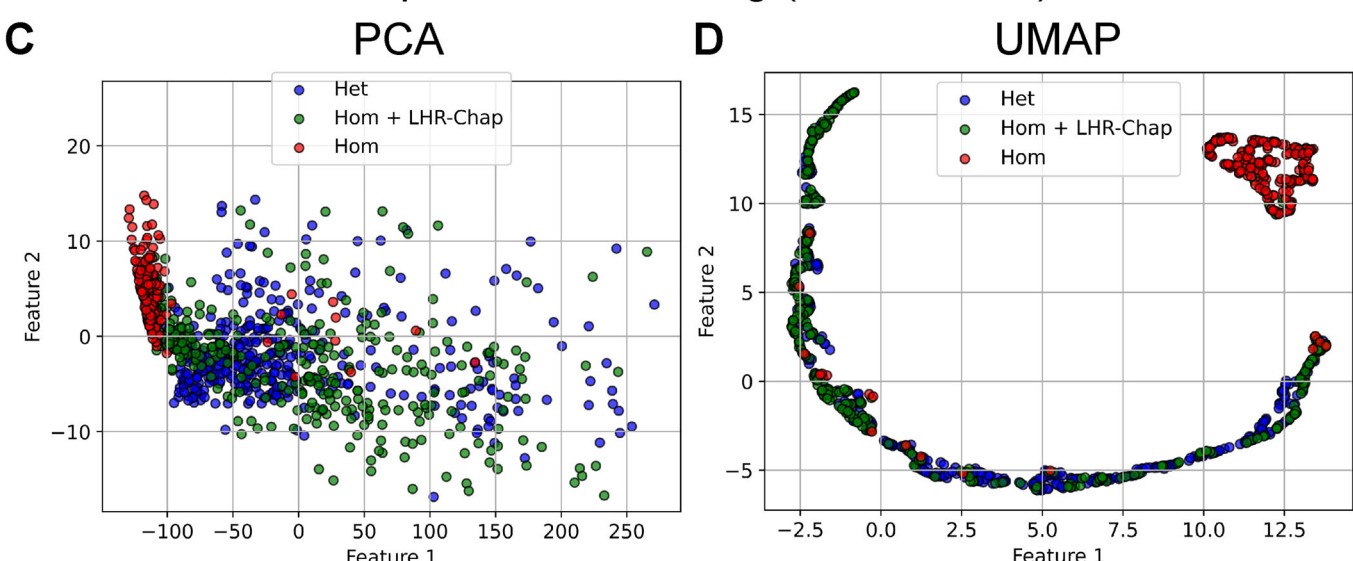

**Figure EV5.  PCA and UMAP plots for visualizing the distribution of ovarian cells based on their calcium profiles.**

(A) A PCA plot of unsupervised clustering (without AI) showed mutant Hom cells being restricted largely to one side of the plot, although with considerable overlap with both control Het and LHR-Chap-treated Hom ovarian cells, which were both more widely dispersed. Beyond this, no other meaningful subclusters could be detected. (B) A UMAP plot showed a more restricted distribution of cells from all three groups, with small subclusters emerging. A minor overlap was observed between the Hom and LHR-Chap-treated Hom cells, which otherwise clustered closer to the control Het cells. (C) PCA plot of AI-based supervised clustering shows a much tighter clustering of the Hom cells, overlapping very little with cells from the other two groups; control Het and LHR-Chap-treated Hom ovarian cells were distributed close to each other, with few distinct clusters. (D) UMAP representation of supervised clustering revealed a tight cluster of Hom cells, which were well-separated from control and treated cells. Subclusters composed mostly of either control or treated cells indicate that the treatment does not restore the mutant cells to exact control conditions, achieving imperfect functional rescue that approximates the calcium profiles of the control cells.

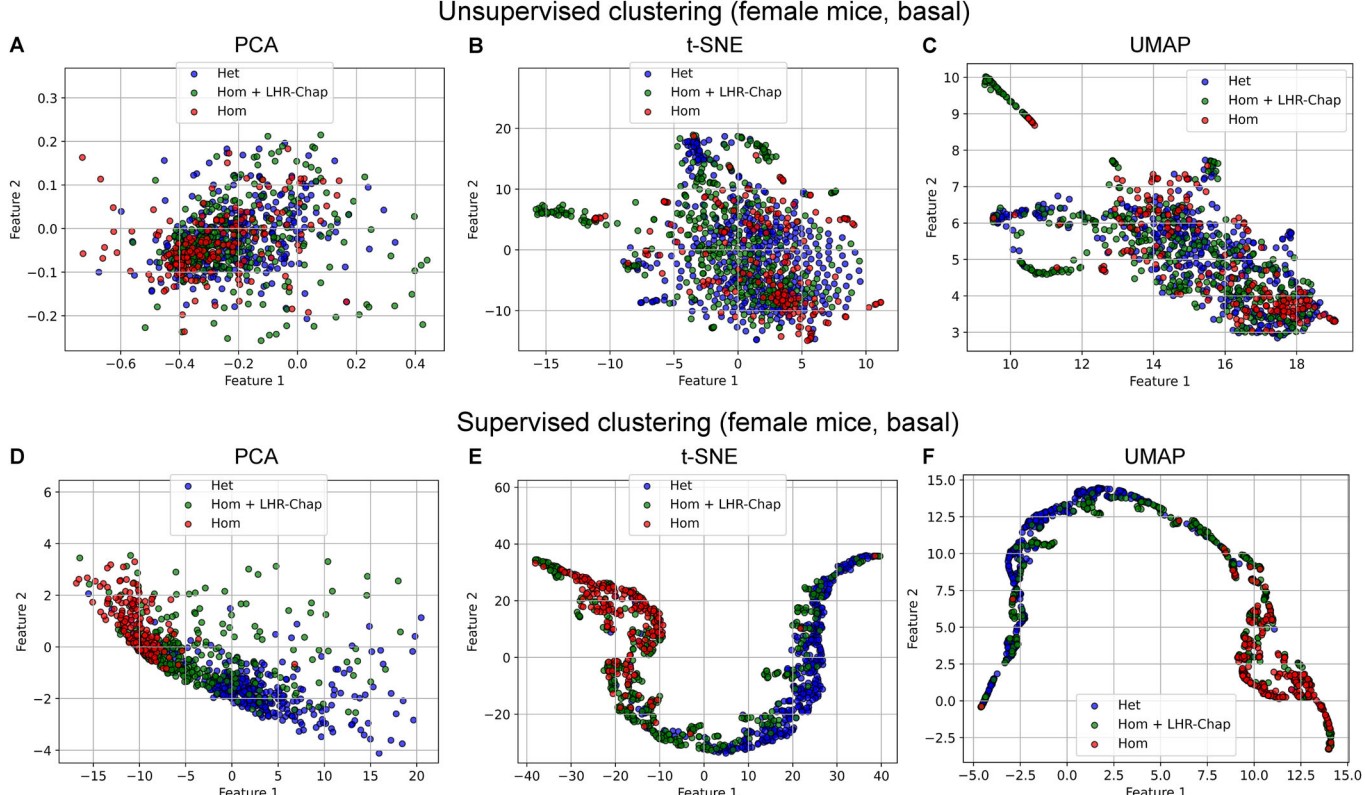

**Figure EV6. AI can distinguish Het from Hom cells, and assess functional impact of treatment, based on spontaneous (basal) calcium profiles alone.**

(A–C) Visualization of unsupervised clustering of ovarian cells based on spontaneous calcium signals (corresponding to "basal" section of calcium imaging) using PCA (A), t-SNE (B), and UMAP (C) approaches did not reveal meaningful clusters. (D) Supervised (AI model) clustering based on spontaneous calcium signals revealed a clearly skewed distribution pattern, even with PCA. (E, F) t-SNE (E) and UMAP (F) visualization show a clear segregation of untreated mutant Hom cells from control Het and LHR-Chap-treated Hom cells, showcasing the ability of the feature-learning model to tease apart these groups based on distinctive spontaneous calcium signaling patterns alone.

