## [Peer Review File · EMBO Molecular Medicine]

Functional rescue and AI analysis of a human inactivating GPCR mutation using a small molecule

Debajyoti Das, Amanda Wyatt, Sarath Sivaprasad, Vanessa Wahl, Sen Qiao, Fabien Ectors, Zulfiah Moosa, Claire Newton, Mario Fritz, Robert Millar, and Ulrich Boehm

Corresponding authors: Ulrich Boehm (ulrich.boehm@uks.eu) , Robert Millar (bob.millar@up.ac.za)

Review Timeline:

Submission Date:	16th Apr 25
Editorial Decision:	3rd Jun 25
Revision Received:	15th Oct 25
Editorial Decision:	26th Nov 25
Revision Received:	8th Dec 25
Accepted:	11th Dec 25

Editor: Lise Roth

Transaction Report:

3rd Jun 2025

Dear Prof. Boehm,

Thank you for submitting your manuscript to EMBO Molecular Medicine, and please accept my apologies for the delay in getting back to you, as securing referees with the adequate expertise necessitated more time than usual, and one referee additionally requested a short extension to provide their report.

We have now received feedback from the three referees who agreed to evaluate your manuscript. As you will see from the reports below, while referee #1 mentions the limited conceptual advance in light of previous studies, referees #2 and #3 are overall supportive of the study.

After further discussion within the team, we find that the study presents sufficient conceptual novelty to be suitable for EMM, and we would therefore like to invite major revisions of the work that would address all experimental concerns. Novelty over previous work should be discussed.

EMBO Molecular Medicine only allows a single round of revisions, and acceptance or rejection of the manuscript will depend on how complete your responses are in the final version. If you would like to discuss further the points raised by the referees, I am available to do so via email or video. Let me know if you are interested in this option.

We are expecting your revised manuscript within three to four months, if you anticipate any delay, please contact us.

We require:

4) A .docx formatted letter INCLUDING the reviewers' reports and your detailed point-by-point responses to their comments. As part of the EMBO Press transparent editorial process, the point-by-point response is part of the Review Process File (RPF), which will be published alongside your paper.

5) A complete author checklist, which you can download from our author guidelines (<https://www.embopress.org/page/journal/17574684/authorguide#submissionofrevisions>). Please insert information in the checklist that is also reflected in the manuscript. The completed author checklist will also be part of the RPF.

6) All Materials and Methods need to be described in the main text using our 'Structured Methods' format. According to this format, the Methods section includes a Reagents and Tools Table (listing key reagents, experimental models, software and relevant equipment and including their sources and relevant identifiers) followed by a Methods and Protocols section describing the methods, ideally using a step-by-step protocol format. The aim is to facilitate adoption of the methodologies across labs. Please download and fill our Reagents and Tools Table template (.docx), which you can find in our author guidelines:

7) Please note that all corresponding authors are required to supply an ORCID ID for their name upon submission of a revised manuscript.

8) It is mandatory to include a 'Data Availability' section after the Materials and Methods. Before submitting your revision, primary datasets produced in this study need to be deposited in an appropriate public database, and the accession numbers and database listed under 'Data Availability'. Please remember to provide a reviewer password if the datasets are not yet public (see <https://www.embopress.org/page/journal/17574684/authorguide#dataavailability>).

9) For data quantification: please specify the name of the statistical test used to generate error bars and P values, the number (n) of independent experiments (specify technical or biological replicates) underlying each data point and the test used to calculate p-values in each figure legend. The figure legends should contain a basic description of n, P and the test applied. Graphs must include a description of the bars and the error bars (s.d., s.e.m.). Please provide exact p values.

10) Our journal encourages inclusion of *data citations in the reference list* to directly cite datasets that were re-used and obtained from public databases. Data citations in the article text are distinct from normal bibliographical citations and should directly link to the database records from which the data can be accessed. In the main text, data citations are formatted as follows: "Data ref: Smith et al, 2001" or "Data ref: NCBI Sequence Read Archive PRJNA342805, 2017". In the Reference list, data citations must be labeled with "[DATASET]". A data reference must provide the database name, accession number/identifiers and a resolvable link to the landing page from which the data can be accessed at the end of the reference. Further instructions are available at .

11) We replaced Supplementary Information with Expanded View (EV) Figures and Tables that are collapsible/expandable online. EV Figures should be cited as 'Figure EV1, Figure EV2' etc... in the text and their respective legends should be included in the main text after the legends of regular figures.

12) The paper explained: EMBO Molecular Medicine articles are accompanied by a summary of the articles to emphasize the major findings in the paper and their medical implications for the non-specialist reader. Please provide a draft summary of your article highlighting

13) Author contributions: CRediT has replaced the traditional author contributions section because it offers a systematic machine readable author contributions format that allows for more effective research assessment. Please remove the Authors Contributions from the manuscript and use the free text boxes beneath each contributing author's name in our system to add specific details on the author's contribution. More information is available in our guide to authors.

Please also suggest a visual abstract to illustrate your article as a PNG file 550 px wide x 300-600 px high. A cropped portion of this image will serve as thumbnail for the table of content on our webpage.

16) As part of the EMBO Publications transparent editorial process initiative (see our Editorial at <http://embomolmed.embopress.org/content/2/9/329>), EMBO Molecular Medicine will publish online a Review Process File (RPF) to accompany accepted manuscripts.

In the event of acceptance, this file will be published in conjunction with your paper and will include the anonymous referee reports, your point-by-point response and all pertinent correspondence relating to the manuscript. Let us know whether you agree with the publication of the RPF and as here, if you want to remove or not any figures from it prior to publication. Please note that the Authors checklist will be published at the end of the RPF.

EMBO Molecular Medicine has a "scooping protection" policy, whereby similar findings that are published by others during review or revision are not a criterion for rejection. Should you decide to submit a revised version, I do ask that you get in touch

after three months if you have not completed it, to update us on the status.

I look forward to receiving your revised manuscript.

Yours sincerely,

Lise Roth

**** Reviewer's comments ****

Referee #1 (Remarks for Author):

In this work, Das et al, address the question of the possibility to rescue a mutated luteinizing hormone receptor (LHR) exhibiting altered trafficking in vivo using pharmacochaperones (PC). They selected the human T461I LHR mutation, identified in a patient with type 1 Leydig cell hypoplasia leading to infertility. The mutant females exhibited irregular estrous cycles, anovulation, abrogation of ovarian LHR signaling, and complete infertility. The authors developed a mouse model to study pharmacochaperone (PC)-induced LHR signaling and functional rescue in primary cells from an intact mouse. They describe the generation of an LhrT465I-IC mouse line carrying the corresponding mouse mutation (i.e., T465I in terminal exon 11) of the Lhcgr gene. The authors demonstrate PC-mediated functional rescue in mice carrying a human mutation. PC treatment of mutant females restored LH signaling and estrous cyclicity, normal ovarian functions such as ovulation and corpus luteum formation.

To characterize treatment efficacy, they developed an AI algorithm that reliably identified intrinsic differences between experimental groups, enabling functional analysis of the treatment effect in vivo. The authors conclude that their data pave the way for the use of molecules targeting GPCRs as PCs to treat various diseases resulting from mutations inactivating GPCR.

They conclude that their pave the way for the use of molecules targeting GPCR as PC to treat various diseases resulting from inactivating mutations in GPCR.

The mutated LHR rescue is convincing in the article. However, rescue of G protein receptor (GPCR) function by pharmacochaperone (PC) in mice has already been demonstrated. See for example:

Ortega JT, Parmar T, Carmena-Bargueño M, Pérez-Sánchez H, Jastrzebska B. Flavonoids improve the stability and function of P23H rhodopsin slowing down the progression of retinitis pigmentosa in mice. *J Neurosci Res.* 2022 Apr;100(4):1063-1083. doi: 10.1002/jnr.25021. Epub 2022 Feb 15. PMID: 35165923; PMCID: PMC9615108.

This work would therefore be of interest for a more specialized journal.

Referee #2 (Comments on Novelty/Model System for Author):

- Model system: The knock-in mouse model is biologically relevant and adequately validated.
- Medical impact: The demonstration of PC-mediated functional rescue of a GPCR in vivo represents a significant conceptual and translational advance.

Referee #2 (Remarks for Author):

This manuscript presents a detailed proof-of-concept preclinical study demonstrating the functional rescue of a misfolded GPCR (luteinizing hormone receptor, LHR) by a pharmacological chaperone (Org43553, an allosteric agonist of LHR). The authors generated a knock-in mouse model carrying the T465I mutation (corresponding to T461I in human) and showed that LHR-Chap could restore LHR function (LH-mediated calcium signaling) and fertility phenotypes, particularly in females. The use of AI tools, including a neural network classifier, t-SNE embeddings, and Grad-CAM, offers interpretable insight into dynamic calcium

signals. The study is well-conceived and addresses an important and underexplored therapeutic opportunity in GPCR pharmacology.

Major comments

1. Imbalanced comparison between AI models

In the unsupervised approach, the original 1500-dimensional calcium trace data are reduced to 2 dimensions using t-SNE. However, t-SNE is primarily optimized for visualization and does not necessarily preserve all information or global structure in the data. The rationale for choosing a 2D projection is not clearly provided. Why was 2D chosen instead of a higher-dimensional embedding, which might retain more of the original signal structure? Moreover, the authors do not justify the choice of t-SNE over other dimensionality reduction techniques such as PCA or UMAP. In contrast, the supervised neural network is trained directly on the full 1500D input, leading to well-separated feature representations. To more robustly support the effectiveness of the neural network, the authors should compare its classification performance with other standard machine learning methods, such as Support Vector Machines (SVM) or Decision Trees, trained on the same 1500D input data.

2. Lack of novelty in AI methodology

The AI-assisted analysis relies on well-established tools (t-SNE, K-means, feedforward neural nets, Grad-CAM) applied in a straightforward pipeline. While integration into a biological system is valuable, there is limited methodological innovation from the AI perspective. The authors should clarify whether the neural network was specifically optimized for time-series data, and explicitly state that the novelty lies in the biological application and not in the machine learning algorithm, to manage expectations.

3. Image analysis: human expert annotation lacks methodological detail.

The manuscript mentions that ROIs were annotated by "a human expert," but this lacks sufficient detail to assess reliability or reproducibility. Please provide more information on the following:

- How were ROIs marked by the human expert?
- Was the expert blinded to experimental group labels to avoid potential bias?
- Was inter-rater variability assessed (e.g., through comparison with a second independent expert)?
- How many images and ROIs were annotated in total?

These clarifications are important, especially since the annotated ROIs are used as ground truth to evaluate the performance of the AI model.

Minor Comments

1. Notation Correction in K-means Algorithm Description (line 150, SI file) " $X = \{x_1, x_2, \dots, x_n\} \in \mathbb{R}^2$ " is mathematically inaccurate. X is a set of vectors, $\{x_1, x_2, \dots, x_n\}$, so it's a collection of n points.

2. Supplementary figure S10: the saliency map should include multiple time points

3. Overstatement in the abstract: The abstract begins by referencing "large libraries" of small-molecule GPCR ligands for repurposing, but only one compound was tested (Org42599/Org43553). The authors should either tone down this generality or indicate this is a proof-of-concept using a known ligand.

4. Time axis units in calcium trace plots (e.g., Figure 5D, 10D, 14C) should be explicitly labelled.

5. Figure 10D top right panel, first calcium trace appears distinct from the rest. The authors should clarify whether these represent outlier cells or technical artifacts.

Referee #3 (Comments on Novelty/Model System for Author):

nice study but could include data analysis on transgenic LHR(WT) CRE mice (described in comments to authors).

Referee #3 (Remarks for Author):

In the comprehensive study by Das et al, the authors describe how mutations in the luteinizing hormone receptor LHR results in both signaling and developmental and functional defects in vivo but can be overcome by pharmacological chaperones (PCs). Using a targeted mouse model the investigators tested the capacity of LHR point mutants, previously described as folding mutants, to elicit defective signaling and development of sex organs in male and female mice. The investigators also generated a novel set of mice by crossing the LHR transgenics with a GCamps sensor in order to measure Ca^{2+} responses in vitro. With these tools in hand the investigators tested the capacity of previously described pharmacological chaperones to restore the defective signaling and developmental defects observed in the mutant LHR mice. In addition the authors incorporated artificial intelligence to interpret the in vitro Ca^{2+} imaging properties of the mutant mice as well as the effect of the PCs. The goal is to provide a novel method to interpret the complex, multifactorial responses observed in hormone signaling using in vitro slice analysis in the mutant LHR mice. Here the authors were able to identify clear but distinct populations of cells in the heterozygous and homozygous mice for the LHR mutants, that respond to luteinizing hormone and/or PCs.

Overall, this is very nice and comprehensive study. It incorporates the description of novel mouse phenotypes depicting both signaling and developmental defects, as well as their rescue by PCs. Moreover the study incorporates the use of AI to interpret the in vitro data that may have broader potential implications to drug development of therapeutics in general.

There are several points the authors need to address to clarify the message in their study. There are potentially new experiments that may be required unless the authors can provide a rationale for their exclusion.

Comments:

The authors need to elaborate how the Org compounds are not merely LHR binding ligand but are actually Ago-PAMs. The fact that they are allosteric modulators has several implications in that their maximal effects in vivo (eg. during development) may be influenced by circulating LH. The fact that the compounds themselves have agonist activity is also important to note readers. This may be one difference between the developmental and in vitro (slice) analysis.

The authors need to either test the Ca²⁺ responses in WT mice (using the same reporter and CRE-LHR(WT) or provide a rationale for excluding these studies. Indeed, some studies have suggested that WT and HET mice appear to be statistically insignificant different (in developmental), the AI analysis might reveal otherwise. This needs to be addressed by the authors either experimentally or provide an explanation for why such studies may not be necessary. Perhaps the authors already have data on the transgenic CRE-LHR(WT) mice that suggest they behave like the HET mice.

The t-distributed stochastic neighbor embedding (t-SNE) analysis was interesting. The authors describe this as a 2D analysis through fluorescence intensity and time as the two dimensions. But isn't this analysis a three-dimensional analysis since the geography or morphology might be the 3rd dimension? The authors should provide some additional or alternate description to delineate the potential confusion.

We thank the reviewers for the thoughtful and constructive comments on our manuscript. Each of these is addressed as described below, either by the addition of data, changes to the manuscript, or explanatory responses. All changes in the revised manuscript are highlighted in blue.

Referee #1 (Remarks for Author):

The mutated LHR rescue is convincing in the article. However, rescue of G protein receptor (GPCR) function by pharmacochaperone (PC) in mice has already been demonstrated. See for example: Ortega JT, Parmar T, Carmena-Bargueño M, Pérez-Sánchez H, Jastrzebska B. Flavonoids improve the stability and function of P23H rhodopsin slowing down the progression of retinitis pigmentosa in mice. J Neurosci Res. 2022 Apr;100(4):1063-1083. doi: 10.1002/jnr.25021. Epub 2022 Feb 15. PMID: 35165923; PMCID: PMC9615108.

This work would therefore be of interest for a more specialized journal.

Response: We respectfully disagree. Rescue of mutant GPCRs is a relatively embryonic therapeutic area with far-reaching possibilities in view of the large number of GPCRs implicated in a wide range of pathologies. While promising *in vitro* results have been reported across several receptor types, *in vivo* proof-of-concept studies remain limited. Our study demonstrates functional rescue of a GPCR by PC in adult females in the most sexually dimorphic physiological system in both mice and humans. To our knowledge, our preclinical model is the first PC-based GPCR rescue to provide genetic access to the primary target cells allowing functional analyses of intracellular signaling upon PC administration in both sexes. Studying direct PC action on target cells is of pivotal importance to tailor future sex-specific pharmacotherapies. Coupled with our demonstration of the immense potential for preclinical classification of the mutant offered by our AI model, we are convinced that our study will be of great interest to the general reader. Furthermore, our study demonstrates the use of an allosteric agonist, which ferries the mutant receptor to the cell surface, and at the same time stimulates the receptor thus avoiding the complex manipulations of removing antagonists and replacing with agonist while still retaining the rescue of receptor to the cell surface. In addition, we have added new data demonstrating that our AI model successfully classifies mutant and control cells based on minute differences in steady-state intracellular calcium profiles alone (new Figure 16 and Table 3), further highlighting its utility as a preclinical classifier.

Referee #2:

Major comments

1.Imbalanced comparison between AI models

In the unsupervised approach, the original 1500-dimensional calcium trace data are reduced to 2 dimensions using t-SNE. However, t-SNE is primarily optimized for visualization and does not necessarily preserve all information or global structure in the data. The rationale for choosing a 2D projection is not clearly provided. Why was 2D chosen instead of a higher-dimensional embedding, which might retain more of the original signal structure? Moreover, the authors do not justify the choice of t-SNE over other dimensionality reduction techniques such as PCA or UMAP. In contrast, the supervised neural network is trained directly on the full 1500D input, leading to well-separated feature representations. To more robustly support the effectiveness of the neural network, the authors should compare its classification performance with other standard machine learning methods, such as Support Vector Machines (SVM) or Decision Trees, trained on the same 1500D input data.

Response: The core reason for selecting AI was to elucidate and interpret patterns in calcium signaling and to demonstrate predictive preclinical classification. We chose a 2D projection with the sole purpose of convenient visualization of the high-dimensional data. All our statistical tests were conducted on the high-dimensional data itself, and the 2D projection was only used for representation. We have added this rationale in the revised manuscript. We have also added full-dimensional (1500D) clustering data in the revised manuscript (new supplementary Tables S2, S4, S6 and S8) to address the reviewer's concern.

We selected t-SNE to represent the high-dimensional data because it preserves local structure the most to reveal clusters (compare new Tables S1, S3, S5 and S7 with Tables S2, S4, S6 and S8, respectively), as demonstrated for biological data (Van der Maaten & Hinton, 2008; Kobak & Berens, 2019). Furthermore, similar techniques are used to reveal relationships between individual cells' profiles, determining which cells behave similarly within complex, heterogeneous populations (Amir et al., 2013). This choice also aligns with our further clustering-based visualization and analyses: t-SNE preserved local clusters in the original data and K Nearest Neighbor clustering helps quantify this clearly. For comparison, we have also added PCA plots (showing highest variance) and UMAP plots (showing global structure of the data) in the revised manuscript (new Figures EV1, EV2 and 16).

As suggested, we compared the performance of our AI-based neural network with that of classical machine learning methods. The purpose of using AI was to develop a feature-learning method that could also visualize the data (as opposed to traditional machine learning methods that allow only classification), capitalizing on the discriminating features among the experimental groups. We also seized the opportunity and trained a deeper AI model with dropout regularization and bolstered the analyses by adding results from 8 different traditional machine learning models in the revised manuscript (new Tables 1, 2, and 3). Our AI model now has 3 layers (revised Figures 6B and 15B; revised Figures S5 and S10 and new Figure 16 and new Tables 1, 2, 3, S3, S4, S7 and S8) with wide hidden layers (512→128), and staged dropout (50% then 20%) to reduce overfitting, making it deeper, wider, and more regularized than the previous model, which typically improves generalization (Böhle et al., 2022).

2. Lack of novelty in AI methodology

The AI-assisted analysis relies on well-established tools (t-SNE, K-means, feedforward neural nets, Grad-CAM) applied in a straightforward pipeline. While integration into a biological system is valuable, there is limited methodological innovation from the AI perspective. The authors should clarify whether the neural network was specifically optimized for time-series data, and explicitly state that the novelty lies in the biological application and not in the machine learning algorithm, to manage expectations.

Response: We agree that the innovation in our study lies primarily in the AI application to biological data and the insights gained therein, demonstrating the potential of AI analysis in this context and explicitly state that the novelty lies in the biological application in the revised manuscript. We have also added new data demonstrating that our AI model successfully classifies mutant and control cells based on minute differences in steady-state intracellular calcium profiles alone (new Figure 16 and Table 3), further highlighting its utility as a preclinical classifier.

We have clarified that the neural network was indeed optimized for time-series data. We have added some initial proposals toward an automated pipeline that helps analyze effects of similar therapeutic interventions (Figure S12) and explainable AI (XAI)-based analysis using faithful explanations obtained from the B-Cos network (revised Figure S13). While we do not present this as methodological innovation, to our knowledge, this is the first application of the B-Cos network, which was developed by us in a previous study (Böhle et al., 2022), to biological functional imaging data. This is clearly stated in the revised manuscript without overstating its novelty.

3. Image analysis: human expert annotation lacks methodological detail.

The manuscript mentions that ROIs were annotated by "a human expert," but this lacks sufficient detail to assess reliability or reproducibility. Please provide more information on the following:

- *How were ROIs marked by the human expert?*
- *Was the expert blinded to experimental group labels to avoid potential bias?*
- *Was inter-rater variability assessed (e.g., through comparison with a second independent expert)?*
- *How many images and ROIs were annotated in total?*

These clarifications are important, especially since the annotated ROIs are used as ground truth to evaluate the performance of the AI model.

Response: The ROIs were marked manually (by hand) using ImageJ. The human expert was blinded to the experimental groups to avoid bias. We also assessed inter-rater variability with a second independent human expert, and the variation was found to be minimal.

A total of 56 time-series confocal recordings (each comprised of 1500 images) were used. A total of 1946 ROIs (cells) were annotated. We have added this information in the Materials and Methods section of the revised manuscript.

Minor Comments

1. Notation Correction in K-means Algorithm Description (line 150, SI file) " $X = \{x_1, x_2, \dots, x_n\} \in \mathbb{R}^2$ " is mathematically inaccurate. X is a set of vectors, $\{x_1, x_2, \dots, x_n\}$, so it's a collection of n points.

Response: Corrected.

2. Supplementary figure S10: the saliency map should include multiple time points

Response: Updated to include multiple time-points.

3. Overstatement in the abstract: The abstract begins by referencing "large libraries" of small-molecule GPCR ligands for repurposing, but only one compound was tested (Org42599/Org43553). The authors should either tone down this generality or indicate this is a proof-of-concept using a known ligand.

Response: We have revised the Abstract to mention that the ligand used is known.

4. Time axis units in calcium trace plots (e.g., Figure 5D, 10D, 14C) should be explicitly labelled.

Response: Time axis units have been labelled at the lower left corner "200 s" (200 seconds) in all calcium trace plots. We have increased the font size to improve legibility. Time axes have also been added for all heatmaps in the revised manuscript.

5. Figure 10D top right panel, first calcium trace appears distinct from the rest. The authors should clarify whether these represent outlier cells or technical artifacts.

Response: The first calcium trace potentially represents a subset of LHR-expressing ovarian cells that are distinct from the rest of the population. We intentionally show this cell to emphasize the apparent heterogeneity in the calcium profiles of LHR cells. We now specifically highlight this in the revised manuscript.

Referee #3:

The authors need to elaborate how the Org compounds are not merely LHR binding ligand but are actually Ago-PAMs. The fact that they are allosteric modulators has several implications in that their maximal effects in vivo (eg. during development) may be influenced by circulating LH. The fact that the compounds themselves have agonist activity is also important to note readers. This may be one difference between the developmental and in vitro (slice) analysis.

Response: We agree and now discuss the allosteric properties of the Org compound and the implications of their effects *in vivo* in the revised version of the manuscript.

The authors need to either test the Ca²⁺ responses in WT mice (using the same reporter and CRE-LHR(WT) or provide a rationale for excluding these studies. Indeed, some studies have suggested that WT and HET mice appear to be statistically insignificant different (in developmental), the AI analysis might reveal otherwise. This needs to be addressed by the authors either experimentally or provide an explanation for why such studies may not be necessary. Perhaps the authors already have data on the transgenic CRE-LHR(WT) mice that suggest they behave like the HET mice.

Response: To address this important point and be able to compare the calcium profiles of WT and Het mice, we generated a novel Cre-independent AAV9-GCaMP3 and microinjected it into the ovaries of WT and Het female mice. Since Cre is not expressed in the WT mouse, this targets all ovarian cells, including the ones expressing LHR. Importantly, we did not find a difference in the basal calcium levels or in the response to LH application between WT and Het groups (new Supplementary Figure S6). We also found that our AI model did not distinguish between cells from these two groups, indicating that ovarian cells from the WT mice behave like those from Het mice in terms of calcium profiles (new Supplementary Figure S11). We have added these new results in the revised manuscript.

The t-distributed stochastic neighbor embedding (t-SNE) analysis was interesting. The authors describe this as a 2D analysis through fluorescence intensity and time as the two dimensions. But isn't this analysis a three-dimensional analysis since the geography or morphology might be the 3rd dimension? The authors should provide some additional or alternate description to delineate the potential confusion.

Response: In the present study, only the calcium imaging heatmaps show change in fluorescence intensity with time (Figures 5A, 5B, 10A, 10B, 14A, and S6A, S6B). For the t-SNE-based analysis, each cell is represented solely by its 1500-dimensional calcium signal. Therefore, each cell is identified by a unique high-dimensional “functional signature” that contains no spatial, morphological, or geographical co-ordinates or dimensions. The t-SNE algorithm projects these high-dimensional time-series vectors into a two-dimensional abstract space whose axes have no direct biological meaning; they exist only to arrange cells so that those with similar calcium-signaling dynamics lie near one another. Because neither tissue geography nor cellular morphology is supplied to the algorithm, no third spatial dimension is involved, and the analysis is described as two-dimensional.”

To make this point clearer, we have added the following text to the Methods section of the revised manuscript: “Each cell’s unique time-series calcium signature ($\Delta F/F_0$ over 1500 time points) was treated as a high-dimensional vector in R^{1500} , with each time-point representing one dimension (1500 dimensions in total). These vectors were reduced using t-SNE (and other classical machine learning methods) to a 2D embedding that groups cells according to similarity in their calcium-signaling dynamics alone; spatial location and morphology were not included in this analysis.”

26th Nov 2025

Dear Prof. Boehm,

Thank you for submitting your revised study, and please accept my apologies for the delay in getting back to you as one referee needed more time to complete their report. We have now received the reports from referees #2 and #3, and as you will see below, they are overall supportive of publication pending minor revisions. I will therefore be able to accept your manuscript once the following concerns are addressed:

1/ Please address the comments from the referees in your rebuttal letter and/or in the manuscript.

2/ Manuscript text:

- Please remove the blue font, and only indicate in track changes mode any new modification in the text.
- Please correct the order and headings of the sections in the manuscript text to: Abstract / Keywords / The Paper Explained / Introduction / Results / Discussion / Methods / Data Availability / Acknowledgements / Disclosure and Competing Interests Statement / References / Main Figure Legends / Tables / Expanded View Figure Legends.
- Materials and methods should be renamed Methods:
 - o Please include a section on cells (HEK 293T), including origin and testing for mycoplasma contamination. Please also include the cells in your Reagent table.
 - o Statistics: please provide a statement on sample size and randomization.
- Data availability: please remove "The datasets generated during and/or analyzed during the current study are available from the corresponding authors on reasonable request." As per our policy, all primary datasets produced in this study need to be deposited in an appropriate public database, and the accession numbers and database listed under 'Data Availability'. Please provide URLs to access the data.
- Please rename 'Disclosure statement' to 'Disclosure statement and competing interests'.
- Grants and funding should be merged with Acknowledgements.

3/ Figures:

- Please merge some of your figures to reduce the total number of figures to 8-10 figures. You may also increase the number of EV figures to ~8 if you wish. Make sure to update the callouts in the manuscript text accordingly.
- Appendix: please remove the blue font and rename the figures and tables "Appendix Figure S1" etc. and "Appendix Table S1" etc. in the appendix legends. Please upload the final version of the file in PDF format.
- Please address the queries from our data editors in the figure legends:
 1. Please note that the exact p values are not provided in the legends of figures 3A, B; 5C, 14 B, D
 2. Please note that the box plots need to be defined in terms of minima, maxima, centre, bounds of box and whiskers, and percentile in the legends of figures 5C, 10C, 14 B, D
 3. Please note that information related to n is missing in the legend of figure 10 C

4/ Checklist:

- Please complete the information on the top left corner
- Please fill the full section: "Experimental study design and statistics"

5/ Thank you for providing a nice visual abstract. I have cropped a small portion of this image to serve as thumbnail for the table of content on our webpage (attached), please let us know if you agree.

6/ As part of the EMBO Publications transparent editorial process initiative (see our Editorial at <http://embomolmed.embopress.org/content/2/9/329>), EMBO Molecular Medicine will publish online a Review Process File (RPF) to accompany accepted manuscripts.

This file will be published in conjunction with your paper and will include the anonymous referee reports, your point-by-point response and all pertinent correspondence relating to the manuscript. Let us know whether you agree with the publication of the RPF.

I look forward to receiving your revised manuscript.

Yours sincerely,

Lise Roth

***** Reviewer's comments *****

Referee #2 (Remarks for Author):

The revised manuscript demonstrates noticeable improvements and resolves several earlier concerns regarding the AI-based analytics. However, it is important to note that the input data used for the AI models remain relatively simple, and the analytical approach lacks methodological novelty, as it primarily involves the application of standard algorithms for data classification and visualization. Although the reported results suggest that the neural network model achieves superior performance, the comparison between AI models is potentially biased because the parameters for the alternative methods were manually selected and kept fixed. The outcomes may differ if the authors conduct a systematic parameter search or optimisation procedure to identify the best-performing configurations for each algorithm.

Referee #3 (Comments on Novelty/Model System for Author):

This is the novel use of AI to assist in analysis of the effects of an allosteric modulator to rescue the behavior of a GPCR with a mutant allele that leads to impaired organ development

Referee #3 (Remarks for Author):

The revised manuscript by Das et al, titled "Functional rescue and AI analysis of a human inactivating GPCR mutation using a small molecule" represents a significant improvement over the previous submission. For the most part the authors addressed this reviewer's comments, with the exception of query 1. In the revised manuscript the authors did distinguish this study over previous literature by the use of an allosteric modulator whereas other pharmacological chaperones were orthosteric ligands. However I felt the authors missed the point of describing the behavior of this particular allosteric modulator since it has Ago-PAM properties. In addition to its positive allosteric effect on agonist like LH, this allosteric modulator has agonist properties, on its own, albeit with lower efficacy than a full orthosteric agonist. Here, the behavior of the Org compound may be different in vivo where circulating LH may cooperatively enhance its activity, compared to in vitro studies (without exogenous LH). Thus, AgoPAMs would exhibit high efficacy in vivo but low efficacy in vitro. This is the point that the authors did not sufficiently describe as it pertains to the actions of the Org compounds both in vivo (developmental) and in vitro studies.

96 *Referee #2 (Remarks for Author):*

97
98 *The revised manuscript demonstrates noticeable improvements and resolves several earlier concerns*
99 *regarding the AI-based analytics. However, it is important to note that the input data used for the AI*
100 *models remain relatively simple, and the analytical approach lacks methodological novelty, as it primarily*
101 *involves the application of standard algorithms for data classification and visualization. Although the*
102 *reported results suggest that the neural network model achieves superior performance, the comparison*
103 *between AI models is potentially biased because the parameters for the alternative methods were*
104 *manually selected and kept fixed. The outcomes may differ if the authors conduct a systematic parameter*
105 *search or optimisation procedure to identify the best-performing configurations for each algorithm.*

106 **Response:** We have added the following sentences to the revised manuscript to address the concerns
107 raised by this reviewer.

108 Although the AI tools used here are themselves not methodologically novel, the novelty of this study lies
109 in demonstrating their utility as a preclinical classifier, i.e. classification of control, mutant, and rescued
110 primary cells from a preclinical mouse model with a human inactivating GPCR mutation, based on their
111 intracellular calcium profiles. In future work, the performances of the AI/ML models could potentially be
112 further enhanced by conducting a systematic parameter search or optimization procedure to identify the
113 best-performing configurations for each algorithm.

114

115 *Referee #3 (Comments on Novelty/Model System for Author):*

116

117 *This is the novel use of AI to assist in analysis of the effects of an allosteric modulator to rescue the*
118 *behavior of a GPCR with a mutant allele that leads to impaired organ development*

119

120 *Referee #3 (Remarks for Author):*

121

122 *The revised manuscript by Das et al, titled "Functional rescue and AI analysis of a human inactivating*
123 *GPCR mutation using a small molecule" represents a significant improvement over the previous*
124 *submission. For the most part the authors addressed this reviewer's comments, with the exception of query*
125 *1. In the revised manuscript the authors did distinguish this study over previous literature by the use of an*
126 *allosteric modulator whereas other pharmacological chaperones were orthosteric ligands. However I felt*
127 *the authors missed the point of describing the behavior of this particular allosteric modulator since it has*
128 *Ago-PAM properties. In addition to its positive allosteric effect on agonist like LH, this allosteric*
129 *modulator has agonist properties, on its own, albeit with lower efficacy than a full orthosteric agonist.*
130 *Here, the behavior of the Org compound may be different in vivo where circulating LH may cooperatively*
131 *enhance its activity, compared to in vitro studies (without exogenous LH). Thus, AgoPAMs would exhibit*
132 *high efficacy in vivo but low efficacy in vitro. This is the point that the authors did not sufficiently describe*
133 *as it pertains to the actions of the Org compounds both in vivo (developmental) and in vitro studies.*

134 **Response:** As requested, we have added new text in the revised manuscript emphasizing the effects of
135 LHR-Chap due to its Ago-PAM properties:

136 LHR-Chap is an agonistic positive allosteric modulator (Ago-PAM). In addition to its positive allosteric
137 effect on the action of the orthosteric (endogenous) agonist, LH (Newton et al., 2011), an Ago-PAM also
138 has agonistic properties. This agonistic activity of LHR-Chap has been shown to induce ovulation (van de
139 Lagemaat et al., 2009), albeit with lower efficacy than LH. Therefore, LHR-Chap may cooperatively
140 enhance the activity of endogenous circulating LH, leading to overall higher efficacy *in vivo* than that
141 observed *in vitro* (without exogenous LH).

142

11th Dec 2025

Dear Prof. Boehm,

Thank you for submitting your revised files. I am pleased to inform you that your manuscript is accepted for publication and is now being sent to our publisher to be included in the next available issue of EMBO Molecular Medicine!

You may qualify for financial assistance for your publication charges - either via a Springer Nature fully open access agreement or an EMBO initiative. Check your eligibility: <https://link.springer.com/journal/44321/how-to-publish-with-us>

Yours sincerely,

Lise Roth

>>> Please note that it is EMBO Molecular Medicine policy for the transcript of the editorial process (containing referee reports and your response letter) to be published as an online supplement to each paper. If you do NOT want this, you will need to inform the Editorial Office via email immediately. More information is available here: <https://link.springer.com/partners/embo-press/editorial-policies#Peer%20review>